# HIV restriction factor APOBEC3G binds in multiple steps and conformations to search and deaminate single-stranded DNA

**Michael Morse[1], M Nabuan Naufer[1], Yuqing Feng[2], Linda Chelico[2], Ioulia Rouzina[3], Mark C Williams[1]\***

[1]Department of Physics, Northeastern University, Boston, United States; [2]Department of Biochemistry, Microbiology and Immunology, University of Saskatchewan, Saskatoon, Canada; [3]Department of Chemistry and Biochemistry, Ohio State University, Columbus, United States

**Abstract** APOBEC3G (A3G), an enzyme expressed in primates with the potential to inhibit human immunodeficiency virus type 1 (HIV-1) infectivity, is a single-stranded DNA (ssDNA) deoxycytidine deaminase with two domains, a catalytically active, weakly ssDNA binding C-terminal domain (CTD) and a catalytically inactive, strongly ssDNA binding N-terminal domain (NTD). Using optical tweezers, we measure A3G binding a single, long ssDNA substrate under various applied forces to characterize the binding interaction. A3G binds ssDNA in multiple steps and in two distinct conformations, distinguished by degree of ssDNA contraction. A3G stabilizes formation of ssDNA loops, an ability inhibited by A3G oligomerization. Our data suggests A3G securely binds ssDNA through the NTD, while the CTD samples and potentially deaminates the substrate. Oligomerization of A3G stabilizes ssDNA binding but inhibits the CTD's search function. These processes explain A3G's ability to efficiently deaminate numerous sites across a 10,000 base viral genome during the reverse transcription process.

**\*For correspondence:**
ma.williams@northeastern.edu

**Competing interests:** The authors declare that no competing interests exist.

## Introduction

The APOBEC3 family of proteins provides humans and other primates with innate immune antiviral resistance. APOBEC3G (A3G) is particularly known for its ability to inhibit the infectivity of human immunodeficiency virus type 1 (HIV-1) (*Harris and Liddament, 2004*; *Malim, 2009*) when it is not targeted for degradation by the accessory protein viral infectivity factor (Vif) (*Desimmie et al., 2014*; *Fisher et al., 1987*; *Goila-Gaur and Strebel, 2008*; *Sheehy et al., 2002*; *Strebel et al., 1987*). A3G primarily functions as a cytidine deaminase (*Suspène et al., 2004*; *Yu et al., 2004*), catalyzing the conversion of deoxycytidine to deoxyuridine on single-stranded DNA (ssDNA). A3G-mediated deamination can result in DNA degradation through cleavage due to the presence of U bases in the viral DNA or G to A base mutations in the complementary DNA, impairing viral replication (*Lecossier et al., 2003*; *Mangeat et al., 2003*; *Pollpeter et al., 2018*; *Zhang et al., 2003*). Structurally, A3G consists of two zinc coordinating domains. Other members of the APOBEC3 family also have either one or two domains, though for two domain proteins only one domain is enzymatically active (*LaRue et al., 2008*). For A3G, the C-terminal domain (CTD) is enzymatically active (*Haché et al., 2005*; *Navarro et al., 2005*), though when isolated it binds ssDNA with low affinity (*Chelico et al., 2010*; *Harjes et al., 2009*). In contrast, the N-terminal domain (NTD), though lacking deaminase activity, binds ssDNA and RNA with high affinity (*Chelico et al., 2010*; *Huthoff et al., 2009*). The NTD is critical in A3G's packaging in the virion and also contains the binding site for Vif

(*Navarro et al., 2005*; *Russell et al., 2009*). A3G also has a self-affinity and oligomerizes, especially when bound to nucleic acids (*Chelico et al., 2008*). A3G oligomerization on ssDNA greatly slows dissociation (*Chaurasiya et al., 2014*; *Morse et al., 2017*). The NTD is important for A3G dimerization, and mutation of two aromatic residues within the NTD (F126/W127), causes A3G to remain monomeric, even when bound to ssDNA (*Chelico et al., 2010*; *Morse et al., 2017*).

Despite the importance of A3G deamination activity, it has been shown that even A3G mutants lacking enzymatic function retain a significant degree of anti-viral function (*Gillick et al., 2013*; *Iwatani et al., 2006*; *Luo et al., 2007*; *Newman et al., 2005*), suggesting A3G also has a deamination-independent mode of suppressing HIV-1 infectivity (*Holmes et al., 2007*; *Iwatani et al., 2007*; *Levin et al., 2010*). Different models of deaminase-independent HIV-1 replication inhibition have been suggested, including slowing reverse transcription by either direct binding of HIV-1 reverse transcriptase (RT) or through a roadblock mechanism mediated by ssDNA binding (*Iwatani et al., 2007*). However, there are difficulties with either model. The small number of A3G monomers packaged in the virion (*Xu et al., 2007*), along with the much lower binding affinity with A3G for RT versus ssDNA (*Adolph et al., 2013*; *Pollpeter et al., 2018*), would prevent direct inhibition of all encapsidated RT molecules. On the other hand, A3G's enzymatic activity requires very fast binding kinetics to scan the entire viral genome (*Senavirathne et al., 2012*), which would seem to prevent functioning as an RT roadblock. This latter issue has been potentially resolved, however, as oligomerization drastically alters A3G's binding properties (*Chaurasiya et al., 2014*). As oligomerization occurs, A3G remains bound to ssDNA for longer periods of time, which also drastically slows cycling on and off substrates, which inhibits enzymatic activity (*Morse et al., 2017*). Thus, A3G can act as an efficient enzyme or a stable ssDNA binding protein on different timescales depending on oligomerization state.

Another complicating factor in A3G function is that viral DNA is vulnerable to A3G deamination only temporarily during the reverse transcription (RT) process. The (-)DNA is single-stranded after the viral RNA template originally packaged in the virion, on which the (-)DNA is formed, is degraded by the RNaseH domain of RT (*Adolph et al., 2018*). The synthesis of the complementary (+)DNA, which creates a double-stranded DNA (dsDNA) complex, blocks subsequent catalytic activity as A3G is an ssDNA specific deaminase. Additionally, A3G only inhibits viral replication when packaged in the virion, which requires the functional NTD (*Haché et al., 2005*; *Navarro et al., 2005*). Even for WT A3G in the absence of Vif, only $7 \pm 4$ A3G monomers are packaged in the virion (*Xu et al., 2007*). Thus, individual A3G monomers must be able to efficiently search the 9719 base long HIV-1 genome during this small window of opportunity. A3G has been shown to use facilitated diffusion to speed up the search for target deamination sites (*Chelico et al., 2006*). Positively charged A3G can slide along the negatively charged ssDNA strand in a 1-dimensional manner to travel short distances or move larger distances by either dissociating from the ssDNA, diffusing in 3-dimensions, and rebinding at a new random site, or by directly transferring from one part of the strand to another (*Feng et al., 2014*; *Nowarski et al., 2008*). This process not only speeds up the search for target sites but also allows A3G to deaminate multiple sites during one binding event. The probability that A3G will deaminate a second neighboring site before dissociating is defined as enzyme processivity and varies as a function of the distance between the two target sites and the presence of short segments of dsDNA or RNA/DNA hybrids (*Ara et al., 2014*). The ability to retain some degree of processivity in the presence of these barriers suggest A3G has some mechanism to directly transfer from one segment of the DNA substrate to another without fully dissociating into solution. Similarly, the ability of A3G to perform intersegmental transfer can also be observed in enzymatic cycling assays in which A3G is incubated with ssDNA in increasing concentrations to promote collisions between substrates, although 3-dimensional diffusion appears to predominate (*Adolph et al., 2017*; *Nowarski et al., 2008*). A3G retains processivity even at high DNA concentrations, indicating the enzyme remains bound to a substrate long enough to deaminate two sites before undergoing intersegmental transfer (*Adolph et al., 2017*). However, increasing the total DNA and A3G concentration by a factor of 5 (while keeping their molar ratio constant) also increases the effective deamination rate by a factor of 2, consistent with A3G occasionally performing intersegmental transfer events due to DNA-DNA collisions (*Adolph et al., 2017*).

The exact manner in which A3G binds and interacts with ssDNA is not fully understood. No structures of full length A3G in complex with nucleic acids have been resolved. Presumably, A3G must be able to interact with ssDNA in a variety of modes to accomplish the many different functions

described above. In vitro experiments observing A3G-ssDNA interactions, however, have revealed some key features. The inability of A3G to deaminate sites close to the 3′ end of a ssDNA substrate suggests that the catalytically active binding state for A3G positions the CTD towards the 5′ end, such that the inactive NTD occupies this 'dead zone' (*Chelico et al., 2010*). Additionally, although A3G moves bidirectionally along ssDNA, as it lacks an energy source for directed motion and instead relies on thermally driven diffusion (*Senavirathne et al., 2012*), the enzyme acts processively only while deaminating in the 3′ to 5′ direction along the ssDNA substrate (*Chelico et al., 2006*). AFM studies have also directly observed the binding and oligomerization of A3G to ssDNA under a variety of conditions, including the observation that A3G is able to form tetramers on a 69 nt long segment of ssDNA (*Pan et al., 2018*; *Shlyakhtenko et al., 2011*; *Shlyakhtenko et al., 2012*; *Shlyakhtenko et al., 2013*). This suggests that, at saturation, a single A3G monomer has a binding site size ≤17 nt, much smaller than ~30 nt long enzymatic dead zone at the 3′ end.

In this study, we measure the binding of A3G to a single, long ssDNA binding substrate using optical tweezers. We directly measure the binding and dissociation of A3G to ssDNA under a wide range of applied forces to detail the energy landscape of the A3G-ssDNA interaction. We find that A3G binds ssDNA in two distinct conformations, consistent with the NTD securely binding the substrate over the timescale of 100 s and the CTD transiently interacting with the substrate, physically contracting the ssDNA and allowing the catalytically active domain to search and deaminate target sequences. This binding mechanism also allows A3G to stabilize ssDNA loops, which could aid in A3G's ability to transfer to different regions on or between substrates. We find that loop formation, complete dissociation from the ssDNA substrate, and enzymatic activity are all inhibited by A3G oligomerization on ssDNA. These results suggest that A3G's function cannot be fully understood by modeling the protein as a single unit, but that the dynamics of both the NTD and CTD must be taken into account. These complex behaviors must affect A3G's ability to perform an effective 3-dimensional search of the viral DNA for potential sites of deamination and form stably bound structures that could inhibit the full replication of HIV proviral DNA by RT.

## Results

### Measuring A3G binding to ssDNA

An 8100 nt ssDNA construct was tethered between two beads, as described in Materials and methods. An optical tweezers system was used to simultaneously measure the extension of and tension along the ssDNA. The force versus extension curve (FEC) of the ssDNA (*Figure 1A*) can be accurately modeled by the freely jointed chain (FJC) polymer model (*Smith et al., 1996*)

$$X(F) = L \left[ coth \left( \frac{2pF}{kT} \right) - \frac{kT}{2pF} \right] \left( 1 + \frac{F}{S} \right), \tag{1}$$

where $L$ is the contour length (end to end length of the DNA without bending or elastic stretching), $p$ is the persistence length, and $S$ is the elastic modulus (force required to double the extended length of the DNA through elastic stretching). Fitting the FJC model to multiple ssDNA molecules returns average parameters of $L = 0.565 \pm 0.003$ nm, $p = 0.717 \pm 0.024$ nm, and $S = 804 \pm 36$ pN (*Figure 1—figure supplement 1*), consistent with previous measurements under similar conditions (*Smith et al., 1996*). The ssDNA was first stretched in protein-free buffer until a set force was reached, ranging from 10 to 80 pN. While this constant force was maintained, A3G at a concentration of 50 nM was flowed into the sample such that local A3G concentration around the trapped ssDNA changed suddenly (<1 s) in a stepwise manner. As A3G bound the ssDNA, its extension was slightly reduced while the optical tweezers were used to maintain a constant tension (*Figure 1B*), consistent with previous measurements of A3G binding force melted dsDNA (*Chaurasiya et al., 2014*; *Morse et al., 2017*). After 100 s, the free A3G was removed from the sample by flowing in protein-free buffer, allowing bound A3G to dissociate without replacement. However, wild type (WT) A3G quickly oligomerizes, and we did not observe a measurable amount of A3G dissociation at low forces (<50 pN) over the timescale of our experiments. We repeated the experiment with an oligomerization-deficient A3G mutant, F126A/W127A (FW). This mutant, which retains enzymatic activity (*Chelico et al., 2010*) and binds ssDNA with a similar affinity as the WT A3G (*Morse et al., 2017*), has been shown to maintain a primarily monomeric state, even when bound to ssDNA. Thus,

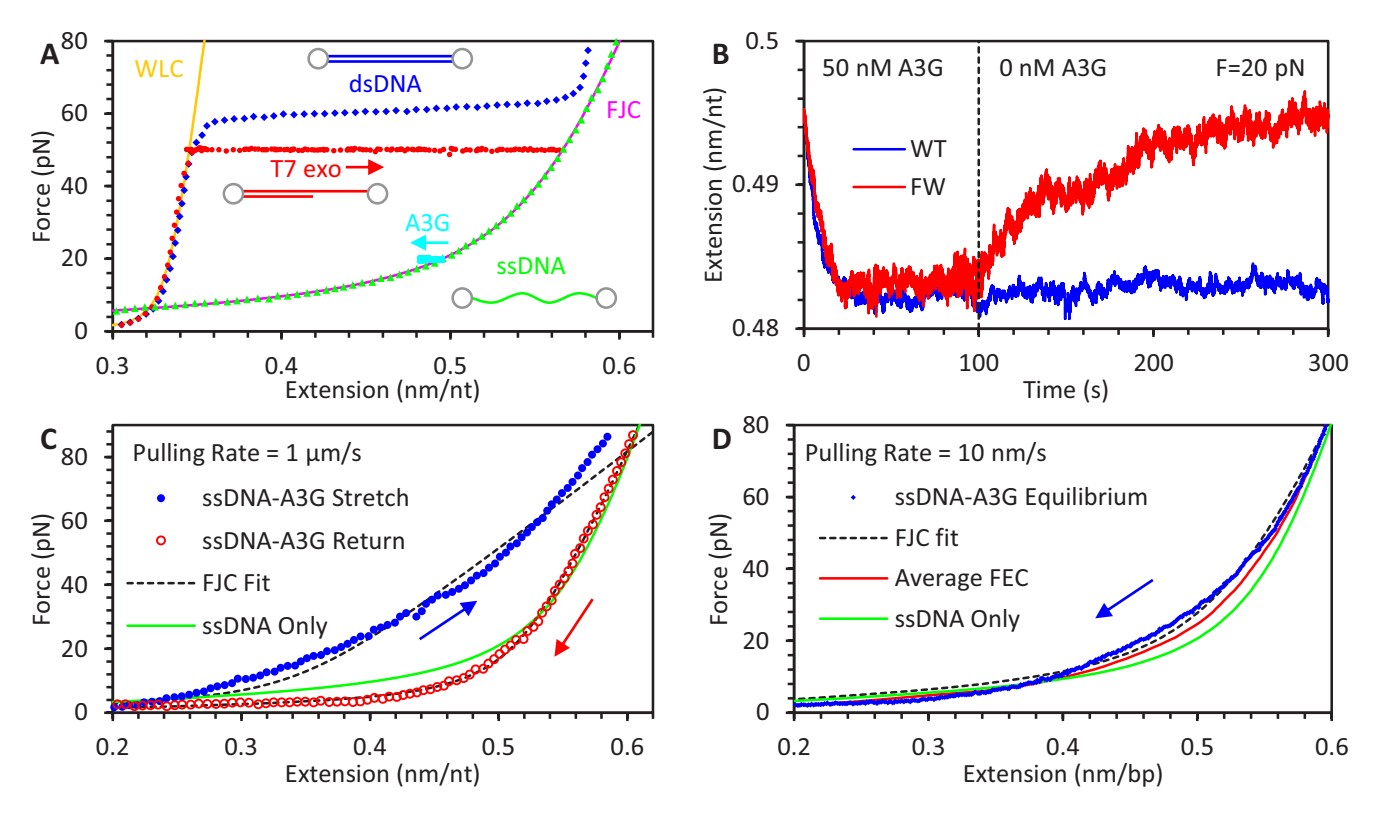

**Figure 1.** Experimental procedure for measuring A3G binding to ssDNA. (**A**) An 8.1 kbp dsDNA template is tethered between two beads, allowing an optical tweezers system to set the DNA's extension and measure the resulting applied force (blue diamonds). The exonuclease activity of T7 DNA polymerase removes one of the strands at high force (red circles), leaving an ssDNA template (green triangles). The FEC of dsDNA and ssDNA can be described by the WLC (yellow line) and FJC (magenta line) polymer models, respectively. The ssDNA is held at a fixed force (cyan squares at 20 pN) and incubated with 50 nM A3G, resulting in a decrease in ssDNA extension. (**B**) Extension of ssDNA while held at constant force (shown at 20 pN) is plotted as a function of time during incubation with 50 nM A3G (WT in blue, FW mutant in red). After 100 s of incubation, the A3G is washed away with protein free buffer. WT A3G (blue line) does not dissociate over the 200 s observation time. When ssDNA is incubated instead with the oligomerization-deficient FW A3G mutant, dissociation is observed on a 100 s timescale and the final extended length approaches that of bare ssDNA. (**C**) FECs of ssDNA in 200 nM of A3G. The ssDNA template is first incubated and saturated with A3G at low force. The extension is rapidly increased (1 µm/s) until a force of 80 pN is achieved (blue solid circles). The ssDNA is allowed to re-equilibrate with the A3G at high force before the extension is rapidly decreased (red hollow circles). While the release can be well fit by the FJC model (dashed lines), the stretch curve cannot be fit by any set of parameters. Compared to the FEC of ssDNA only (green line), the stretch FEC shows greatly decreased ssDNA extension for all forces above ~5 pN while the release FEC shows minor extension reduction at high forces and a moderate extension increase at low force. (**D**) FEC curve of ssDNA stretched at a slow rate of 10 nm/s in 200 nM A3G. The ssDNA is initially held at above 80 pN force before the introduction of A3G, then its extension is slowly decreased to allow A3G to maintain an equilibrium binding state (blue diamonds). Multiple equilibrium FECs are obtained and averaged to find the average extension as a function of force (red line) for A3G-saturated ssDNA.

The online version of this article includes the following figure supplement(s) for figure 1:

**Figure supplement 1 .** Averaging FECs.

while the change in ssDNA extension due to the binding of the A3G FW mutant to ssDNA appears similar to that of the WT during incubation, bound monomers dissociate after the free protein is removed and the ssDNA eventually reaches the same extended length as before A3G incubation.

Alternatively, FECs of ssDNA saturated with A3G can be obtained and compared to the behavior of bare ssDNA to determine how A3G affects the polymer properties of ssDNA. We find, however, that the exact shape of the FEC depends on both the force at which the A3G binds to the ssDNA and the rate at which the ssDNA extension is changed (*Figure 1C*). First, the ssDNA is initially held at low extension and tension before introducing 200 nM A3G. When the ssDNA is rapidly stretched (1 µm/s), the resulting FEC shows a large reduction in ssDNA extension. The ssDNA is then allowed to re-equilibrate at high force before is extension is rapidly decreased. This FEC shows only a minor

decrease in ssDNA extension at high forces and increase ssDNA extension at low force. The difference between these curves indicates the manner in which A3G binds ssDNA and alters its polymer properties is somehow different at low and high forces. Interestingly, the FEC for the ssDNA-A3G complex equilibrated at high force can be fit by the FJC model, while no set of parameters can reproduce the FEC that starts at low force. Furthermore, this FJC fit shows that the binding of A3G increases the persistence length of ssDNA to 1.4 ± 0.1 nm, as evidenced by the lowered force plateau at small extensions and the sharper transition to the elastic stretching regime at higher forces. A similar doubling of ssDNA persistence length has been previously observed for the binding of the single domain APOBEC3H (A3H) (*Feng et al., 2018*). The fast pulling rate used here prevents A3G from fully re-equilibrating as the force changes, resulting in large hysteresis between the low force and high force FECs. In contrast, a slow pulling rate will generate an FEC curve in which the ssDNA-A3G complex maintains a force-dependent equilibrium state (*Figure 1D*). The ssDNA is initially held at high force, then in the presence of 200 nM A3G, slowly (10 nm/s) relaxed to low extension. All FEC curves are obtained multiple times, showing consistent behavior (*Figure 1—figure supplement 1*), allowing the average extension of A3G saturated ssDNA to be measured as a function of force.

## Force dependence of A3G binding kinetics

The change in ssDNA extension, while held at a fixed force, during incubation with 50 nM A3G (*Figure 2A and B*) and A3G dissociation in protein-free buffer (*Figure 2D and E*) both display single exponential behavior in which the ssDNA's extension contracts (incubation) or extends (dissociation) at a constant rate before reaching equilibrium:

$$\Delta X(t) = \Delta X_{eq}(1 - e^{-v_{obs}t}) \tag{2}$$

Here $\Delta X_{eq}$ is the equilibrium extension change (relative to protein-free ssDNA extension). The observed exponential rate ($v_{obs}$), which is the timescale required for protein binding and dissociation to reach equilibrium, is the sum of the bimolecular association rate ($k_{on}$) and dissociation rate ($k_{off}$).

$$v_{obs}(F) = ck_{on}(F) + k_{off}(F) \tag{3}$$

This relationship is derived from the solution to the differential equation governing a two state on-off reaction, and we define the rate of A3G binding to be directly proportional to protein concentration, $c$, as we have previously verified (*Chaurasiya et al., 2014*; *Morse et al., 2017*). During the dissociation step, there is no free protein in solution ($c = 0$) and the observed rate ($v_{obs}$) is exactly the rate of dissociation ($k_{off}$). Thus, the observed rate during the incubation step is used to calculate the rate of A3G binding ($k_{on}$), where $c = 50$ nM.

$$k_{on}(F) = \frac{v_{obs}(F) - k_{off}(F)}{c} \tag{4}$$

We repeated these measurements at several fixed forces, ranging from 10 to 80 pN to determine the concentration-independent rates as a function of force. We observe that the rate of A3G binding decreases with force (*Figure 2C*) and can be well estimated within error as an exponential dependence:

$$k_{on}(F) = k_{on}(0)e^{\frac{-d_B F}{kT}} \tag{5}$$

Here, $d_B$ is a characteristic length scale associated with the reduction in ssDNA extension due to A3G binding. Fitting this relationship to the measured $k_{on}$ (*Figure 2C*) yields a zero-force rate $k_{on}(0) = 0.0095 \pm 0.0015$ s$^{-1}$nM$^{-1}$ and length reduction $d_B = 0.084 \pm 0.005$ nm. In contrast, the rate of bound A3G monomer dissociation does not vary with applied force (*Figure 2F*), yielding a single value $k_{off} = 0.0139 \pm 0.0005$ s$^{-1}$.

We also calculate a force-dependent equilibrium dissociation constant for A3G monomer-ssDNA binding:

$$K_d(F) = k_{off}/k_{on}(F) \tag{6}$$

Over the range of forces studied here, $K_d(F)$ varies between 1 and 10 nM. The degree of ssDNA saturation for a given protein concentration c and value of $K_d$ can be calculated:

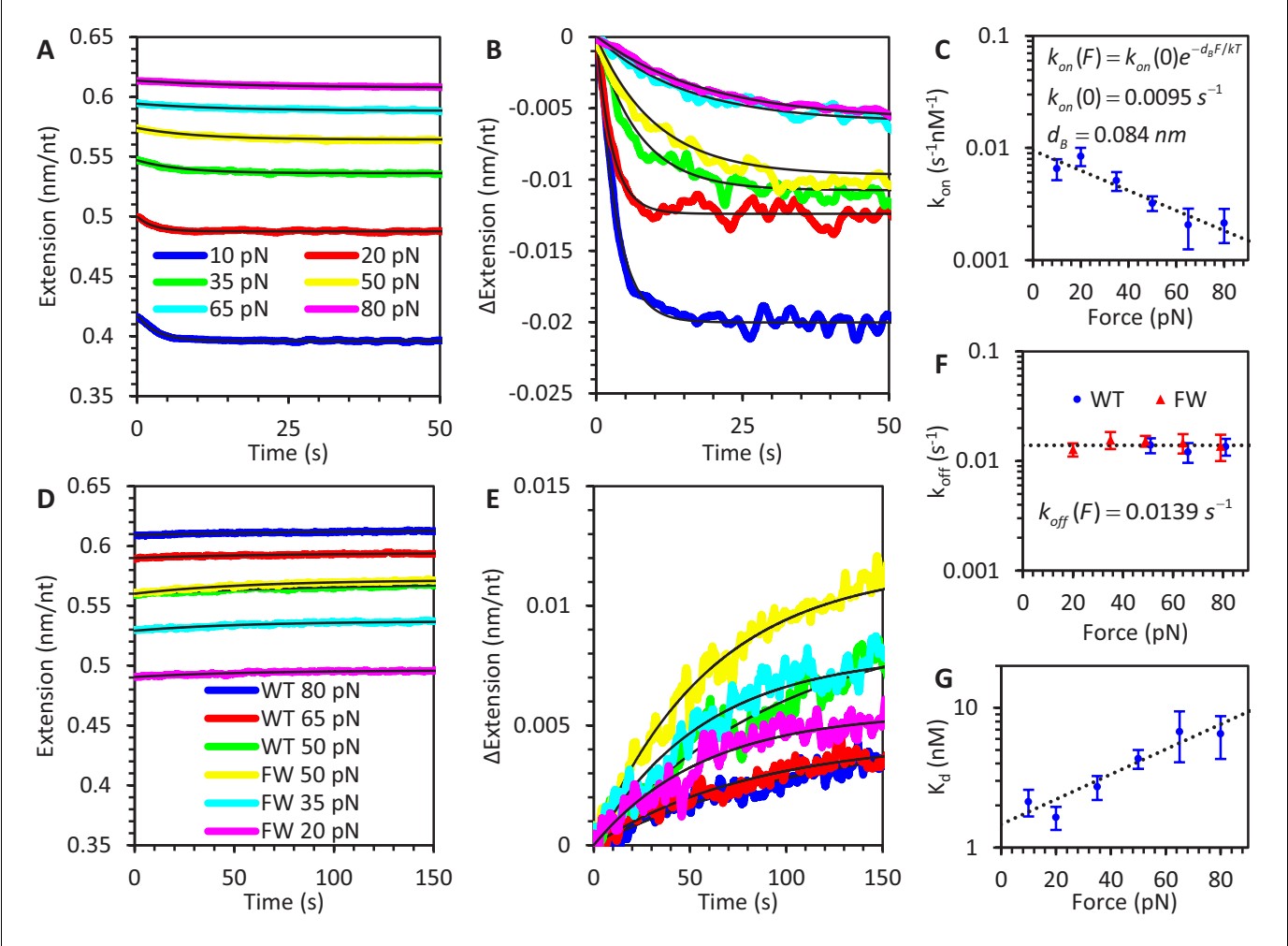

**Figure 2.** Force-dependent kinetics of A3G binding. (A) ssDNA is held at a constant force ranging from 10 to 80 pN and incubated with WT A3G, resulting in a slight reduction in extended length. The initial extension is that of bare ssDNA, as predicted by the FJC model. (B) The same incubation curves are plotted as a net change in extension due to A3G binding, showing that both the rate of binding and equilibrium change in extension decrease as greater forces are applied. (C) Calculated concentration-independent binding rates as a function of force. The rates are fit to an exponential function (dotted line) in which A3G binding is inhibited by increasing force due to a characteristic contraction event $d_B$ upon initial binding. (D) A3G dissociates from ssDNA held at a constant force in the absence of free protein. WT A3G does not dissociate from ssDNA at low forces, but partially dissociates at high forces (F ≥ 50 pN, blue, red, and green lines). Oligomerization-deficient mutant A3G (FW) dissociates over all observed forces (yellow, cyan, and magenta lines). (E) The same incubation curves are plotted as a net change in extension due to A3G dissociation, showing the rate of dissociation is constant with respect to force. (F) Calculated dissociation rates of WT A3G (blue circles) and FW mutant A3G (red triangles), fit by a constant, force-independent value (dotted line. (G) Effective equilibrium dissociation constant as a function of force, calculated by dividing the constant $k_{off}$ value by a force dependent value for $k_{on}$. Error bars are standard error based on multiple experimental replications (N ≥ 5 for $k_{on}$ and N ≥ 3 for $k_{off}$) with different ssDNA molecules.

The online version of this article includes the following source data and figure supplement(s) for figure 2:

**Source data 1.** Numerical values and experimental replicates for data plotted in *Figure 2*.
**Figure supplement 1.** Effect of salt concentration on A3G binding and oligomerization.
**Figure supplement 1—source data 1.** Numerical values and experimental replicates for data plotted in *Figure 2—figure supplement 1*.

$$\theta_{sat} = \frac{c}{c + K_d(F)} \tag{7}$$

This high calculated affinity indicates that even at our highest applied forces, the ssDNA is >85% saturated for the 50 nM A3G constant force experiments and >95% saturated for the 200 nM A3G FEC experiments. Additionally, while nearly all FW mutant A3G dissociates at this rate, a significant

fraction of WT A3G remains bound, especially at low applied forces. Thus, at low forces, the rate of A3G oligomerization is faster than the incubation time (100 s) and dissociation time (~70 s).

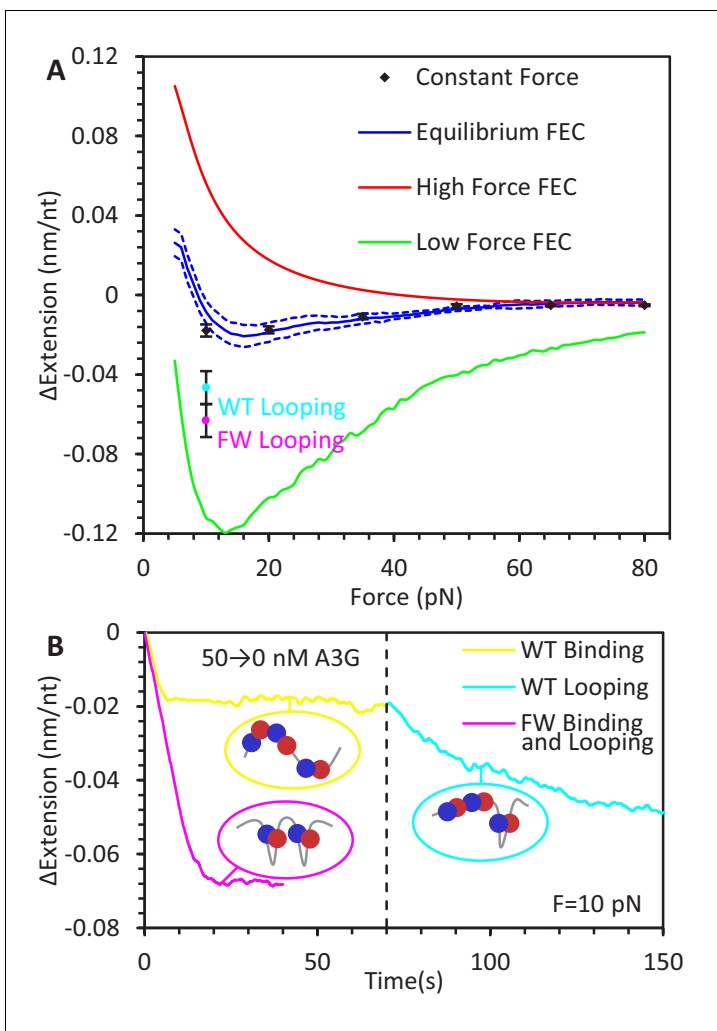

**Figure 3.** Force-dependent change in ssDNA extension due to A3G binding. (**A**) Change in ssDNA extension due to A3G binding as a function of force. Average extension change measured during constant force measurements (black diamonds) increases as force decreases. These measurements agree within error with the force dependent change in extension measured by the averaging the equilibrium FECs (blue line, dotted lines indicate standard error at each force). The changes in extension as measured by the average FEC for A3G bound at high and low forces (red and green lines) are also plotted for comparison, showing the change in ssDNA extension due to saturated A3G binding depends strongly on the force at which the complex is initially equilibrated. (**B**) Additional ssDNA compaction at 10 pN. When A3G is removed from the sample after incubation, the ssDNA undergoes a secondary compaction (cyan line) beyond the initial compaction (yellow line) that occurs in the presence of free A3G. The secondary compaction has a greater amplitude and slower rate than the initial. In contrast, when ssDNA is incubated with FW mutant A3G (magenta line), the initial compaction exhibits a large extension change. The average final extension change due to looping for both WT A3G (cyan point) and FW A3G (magenta point) are plotted in panel A for reference. These results are consistent with A3G binding ssDNA in a compacting conformation, such as the NTD and CTD (blue and red circles) stabilizing variable size ssDNA loops. Error bars are standard error based on multiple experimental replications (N ≥ 5) with different ssDNA molecules.

The online version of this article includes the following source data and figure supplement(s) for figure 3:

**Source data 1.** Numerical values and experimental replicates for data plotted in *Figure 3*.
**Figure supplement 1.** Correcting for A3G saturation.
**Figure supplement 2.** Change in extension for 1x vs 2x ssDNA constructs due to A3G binding at 80 pN.

The majority of experiments presented here used a 50 mM Na$^+$ buffer to allow for direct comparison with previous experiments (*Chaurasiya et al., 2014*; *Morse et al., 2017*). However, higher salt concentrations, including divalent cations, are often present in physiological conditions and in other experimental assays. To check what effect salt concentration can have on A3G binding, we also measured binding of A3G to ssDNA held at 20 pN in both an enzymatic buffer (50 mM Na$^+$ and 10 mM Mg$^+$) and a physiological buffer (150 mM Na$^+$ and 1 mM Mg$^+$) (*Figure 2—figure supplement 1A*). During incubation, we observe similar changes in ssDNA extension due to A3G binding, though the rate of binding is slightly reduced, presumably due to screening of electrostatic interactions (*Figure 2—figure supplement 1B*). Once bound, however, WT A3G quickly oligomerizes under all conditions and does not dissociate once the free protein is removed from the sample. Additionally, when the oligomerization-deficient FW mutant binds the ssDNA, we still observe dissociation of A3G monomers at the same rate as before (*Figure 2—figure supplement 1C*). These results indicate that A3G's abilities to bind and oligomerize are robust over a wide range of salt concentrations and that the behaviors observed in this study should be observed both under physiological conditions and within a wide variety of commonly employed in vitro biochemical assays.

## Variable ssDNA compaction by A3G

The total equilibrated change in ssDNA extension due to A3G binding in our constant force experiments varies greatly with force, ranging from ~0.005 nm/nt at 80 pN to ~0.02 nm/nt at 10 pN (*Figure 3A*, black diamonds). This 4-fold difference cannot be explained by a force-induced decrease in A3G binding affinity, as all experiments result in >85% binding saturation, as previously shown (*Equation 7*). Even directly accounting for any potential decrease in protein saturation results in a correction smaller than standard error of the measurements (*Figure 3—figure supplement 1*). Furthermore, ssDNA extension at each given force obtained from the equilibrium ssDNA FEC obtained in 200 nM A3G corroborates these results (*Figure 3A*, blue line). Instead, the binding of A3G must modify the polymer properties of ssDNA in a complex manner to produce this force-extension relationship.

Three parameters define the polymer properties of ssDNA according to the FJC model (*Equation 1*) and changing any combination of the three will alter the extended length of ssDNA as a function of force. First, we rule out any major changes to the elastic modulus, $S$. While increasing the stiffness of the ssDNA backbone would result in a decrease in extension at a given fixed force, we do not observe any evidence for this in any of the acquired FECs (*Figure 1C and D*), which would present as a steeper slope at high forces in the force-extension plots. Furthermore, an increase in $S$ would result in a larger decrease in extension at high versus low forces, the opposite of the observed trend. Second, while ssDNA extension could be reduced by decreasing persistence length, this value ($p$=0.7 nm) is already much shorter the length scale of A3G itself. Instead, the FJC fit to the high force FEC actually shows evidence that the binding of A3G increases $p$. Physically, this can be interpreted as A3G locally straightening the flexible ssDNA upon binding. An increased value of $p$, however, strictly increases the ssDNA's extension, especially at low forces. The A3G saturated equilibrium FEC does show an extension increase at very low force (<10 pN). This means the primary cause of the observe ssDNA compaction must be a decrease in contour length, $L$. Furthermore, this decrease in $L$ is larger in magnitude when A3G binds ssDNA at low versus high tensions.

The exact manner in which A3G decreases ssDNA contour length is speculative, as no structures of full length A3G in complex with nucleic acid have been resolved. For example, A3G could wrap ssDNA in an organized manner, similar to multidomain single stranded DNA binding proteins, though such a claim would require structural evidence which currently does not exist and would likely result in ssDNA contractions much larger than what we observe. The quantitative analysis performed here, however, is only sensitive to the degree of ssDNA compaction, not the underlying structure. Thus, we simply define that A3G can either bind ssDNA in a non-compacting or a compacting manner. We attempt to determine the source of this compaction, which is inherently inhibited by an opposing applied force along the ssDNA, through the following experiments.

Somewhat counterintuitively, A3G can actually compact ssDNA to a greater degree with reduced saturation. When ssDNA is incubated with A3G at a tension of 10 pN, the initial ssDNA compaction that occurs when free protein enters the sample is followed by a secondary compaction when the free protein is removed (*Figure 3B*). This second extension change is both slower and larger in amplitude than the first. Secondary compaction is not observed at higher applied forces, suggesting

this compaction requires significant slack in the substrate to occur (at 10 pN, ssDNA is only extended to ~70% of its contour length). This suggest A3G has an ability to aggregate flexible ssDNA. Similar aggregation of ssDNA at low forces has also been observed for A3H (*Feng et al., 2018*). This secondary contraction is also observed when we repeat the experiment using the oligomerization-deficient FW mutant A3G, but the entire process occurs at a faster rate and even in the presence of free protein, suggesting this ssDNA aggregation is not the result of A3G oligomerization, but an activity of the monomer itself. One plausible method of accomplishing this function, which we quantitively explore below, is for the two domains of A3G to independently bind the ssDNA substrate. In the absence of a strong straightening force, A3G could stabilize ssDNA loops that are naturally formed as the flexible ssDNA intersects itself.

Previous to this study, the binding of A3G to ssDNA had been observed using optical tweezers by force melting a dsDNA binding substrate (*Chaurasiya et al., 2014*; *Morse et al., 2017*). Under the conditions used (50 mM NaCl, room temperature), the overstretching transition of dsDNA at ~60 pN is primarily due to strand peeling from free ends and internal melting of base pairs (as opposed to the formation base-annealed S-DNA) (*King et al., 2013*). Locally this construct behaves similarly to two ssDNA strands being held under tension, with much of the strands peeled and relaxed, but are joined by short regions of dsDNA that prevent full separation. This construct allows for the binding of a wide variety of ssDNA-specific proteins (*Chaurasiya et al., 2014*; *King et al., 2013*; *Morse et al., 2017*; *Naufer et al., 2016*). As such, we will refer to this substrate as 2x ssDNA (as compared to the single 1x ssDNA strand used in the current experiments). The 2x ssDNA method requires holding DNA continuously at high tension (>60 pN) while A3G binds, preventing measurements at lower forces. We can directly compare, however, these previous results with the 80 pN force measurements performed in this study with 1x ssDNA. While many characteristics of the extension over time curves during A3G binding are quite similar (such as the concentration-dependent rate of binding and the rate of monomer dissociation), the amplitude of the extension change is 10 times larger for the 2x ssDNA template as compared to the 1x ssDNA (*Figure 3—figure supplement 2*). The ~0.05 nm/nt final change in extension upon A3G saturating the 2x ssDNA is approximately equal to the length difference between one ssDNA and two parallel ssDNAs held together at a total tension of 80 pN as predicted by the FJC model. Thus, our data show that the binding of A3G somehow couples or crosslinks the two ssDNA strands together, redistributing the total tension between the strands and increasing the effective stiffness of the 2x ssDNA complex. Again, this behavior is also exhibited by the oligomerization-deficient FW mutant, suggesting the two opposing strands are coupled by individual A3G monomers binding both at the same time. Locally, two strands of melted ssDNA appear very similar to a single flexible ssDNA intersecting itself, suggesting that this interstrand coupling likely performs a similar function as the ssDNA aggregation and compaction discussed above.

## A3G binding conformations

If A3G binds ssDNA in multiple conformation states, and each state impacts the force dependent extension of ssDNA in a different manner, then the application of force should bias the occupancy of these states. This is consistent with our observation that at high forces A3G binds in a manner that increases the ssDNA persistence length but causes a minimal decrease in contour length, but at low forces A3G binds in a manner that greatly reduces contour length. While it is possible the conformation and effective length of the ssDNA-A3G complex changes continuously as applied force deforms the complex, we find that this system is well represented as a simple two state model in which A3G is bound to ssDNA in either a non-compacting or compacting conformation (*Figure 4A*). We show these states graphically as the ssDNA being pinched by the independent binding of the two A3G domains as we propose. We performed a series of force-jump experiments to directly observe transitions between these two conformations. We first allowed an ssDNA template saturated with bound A3G (50 nM) to reach conformational equilibrium at 20 pN, where A3G significantly compacts ssDNA, but we do not observe the secondary compaction that we attribute to larger loop formation (*Figure 3*). Free protein was then removed from the sample so that no additional protein can bind the ssDNA. The extension of the ssDNA was rapidly increased (~1 s) until a set tension ranging from 35 to 80 pN was achieved. At these high forces, ssDNA loops are destabilized, and we observe an overall increase in ssDNA extension (*Figure 4B*). The opposite effect is seen when the A3G is first equilibrated at 80 pN, free protein removed, and then the force decreased to 20 pN or less. The

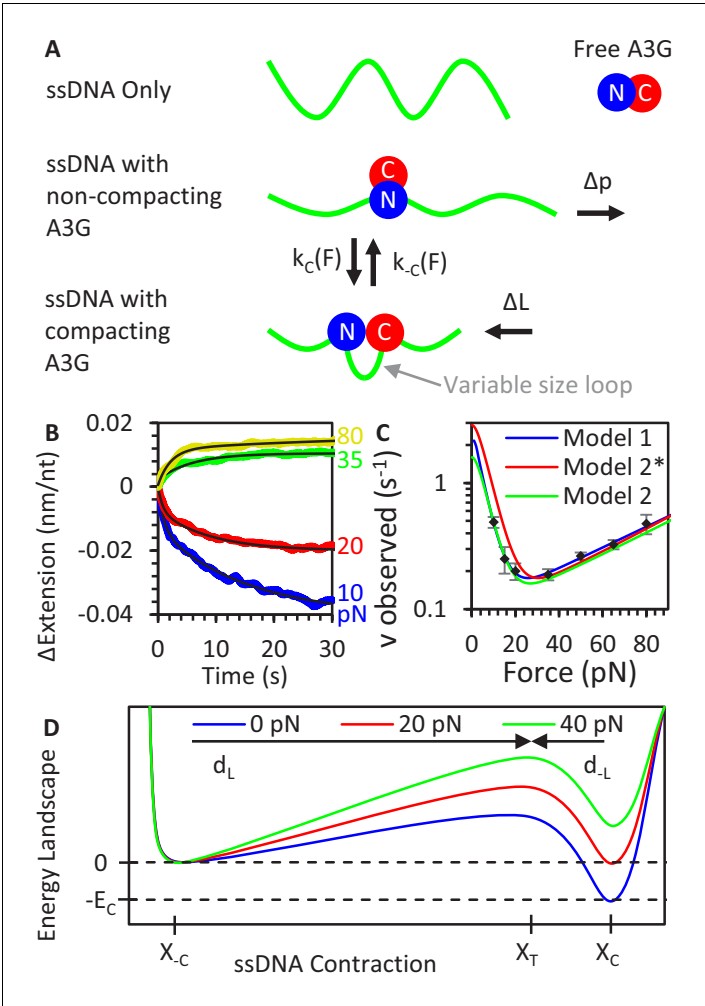

**Figure 4.** A3G binding conformations. (**A**) Schematic of A3G binding conformations. In the absence of strong applied forces, the short persistence length of bare ssDNA results in an extended length much shorter than its contour length. Upon initial binding, A3G locally straightens the ssDNA, increasing its effective persistence length. The A3G can reduce the contour length of the ssDNA by forming a loop of variable size. The degree of looping is determined by the applied tension on the DNA as loops are formed and destabilized at force-dependent rates. (**B**) ssDNA-A3G re-equilibration after force jump. The extension of the ssDNA-A3G complex increases or decreases when the tension on the substrate is suddenly increased or decreased, respectively. The extension as a function of time after the force jump is fit to an exponential function (black lines). (**C**) Force dependence of A3G re-equilibration. The average re-equilibration rates, which is equal to the sum of the loop formation and deformation rates, are plotted as a function of force. All these rates are more than an order of magnitude faster than the measured rate of A3G dissociation (*Figure 2F*), such that this transition is fully attributed to changes in binding conformation of currently bound A3G proteins. (**D**) Energy landscape of A3G conformational change. A3G contracts ssDNA when forming loops, passing through an energy barrier associated with a transition state ($X_T$). At zero force, a binding energy $E_C$ favors the compacting conformation, but increasing applied force increases the energetic cost of compaction. The model is fit to the observed transition rates (panel C, blue line). Using the same parameters as the Bell model fit in a model derived from integrating over deviations between the FECs of ssDNA with and without A3G compaction (panel C, red line) results in slight deviation from the measured rates, but slightly adjusting the parameters recovers the goodness of fit (panel C, green line). Error bars are standard error based on multiple experimental replications ($N \geq 6$) with different ssDNA molecules. Exact parameter values for conformational change models are in *Table 1*.

The online version of this article includes the following source data and figure supplement(s) for figure 4:

**Source data 1.** Numerical values and experimental replicates for data plotted in *Figure 4*.
**Figure supplement 1.** Change in ssDNA extension due to A3G compaction.

extension of the ssDNA decreases over time, consistent with bound A3G transitioning to the more compacting state.

Each change in ssDNA extension over time after a force jump is well fit to a single exponential dependence similar to *Equation 2*, with a notable exception of the 10 pN curves which are fit with two exponential rates. We extract only the faster rate for analysis here, as the slower rate is similar in timescale and amplitude observed during the reorganization of A3G in the absence of free protein seen in the previous experiments (*Figure 3B*). The amplitude of this extension change is nearly equal to the difference in change in extension at the initial and final force (*Figure 3A*), though slightly smaller as the force jump is fast but not instantaneous so a small fraction of the A3G equilibrates during the force ramp. The rate we measure ($v_{observed}$) is the rate at which the bound A3G re-equilibrates between the conformational states and is non-monotonic with force, increasing at both low and high forces (*Figure 4C*). Critically, these measured rates, on the timescale of less than 10 s, are over an order of magnitude faster than the rate of A3G dissociation, so the change in A3G saturation during the experiment is negligible. This 'V' shaped force dependence is characteristic of a two-state system, in which applied force decreases and increases the rate A3G assuming its compacting and non-compacting states, respectively. We first approximate this relationship as a Bell model with each individual A3G monomer switching between two stable states with an exponential dependence on force, such that the observed rate is the sum of the rates of compaction ($k_C$) and decompaction ($k_{-C}$).

$$v_{obs}(F) = k_C(F) + k_{-C}(F) = k_C(0)e^{\frac{-d_C \cdot F}{kT}} + k_{-C}(0)e^{\frac{d_{-C} \cdot F}{kT}} \tag{8}$$

Extrapolated to zero force, the rate of compaction and decompaction are represented as $k_C(0)$ and $k_{-C}(0)$, respectively. The physical significance of $d_C$ is the length that the ssDNA substrate must contract to to allow both domains to bind and $d_{-C}$ is the length that ssDNA must extend to break one of the bounds (we define the signs in *Equation 8* such that both values are positive). This model can be represented as an energy landscape with two potential wells associated with the looped and unlooped conformations separated by a barrier associated with a transition state (*Figure 4D*). In the absence of applied force, the much faster rate of A3G compaction results in this state being greatly favored (~90% occupancy). This implies the compacted state is favored by an energy term $E_L$ on the order of 3 $k_B$T, assuming the two states occupancies can be calculated via Boltzmann distribution. Increased ssDNA tension tilts the energy landscape, making the more contracted looped state less energetically favorable. This model fits the measured data well with values of $d_C$ ~0.8 nm and $d_{-C}$ ~0.08 nm (*Figure 4C*, blue line, exact parameters in *Table 1*). The 10-fold larger $d_C$ compared to $d_{-C}$ value indicates a short-range interaction is responsible for A3G mediated compaction, such that the ssDNA almost fully contracts before both domains can bind while a slight extension of the complex will break this interaction.

We alternatively model the system by explicitly taking into account the polymer properties of the ssDNA substrate being acted upon by the binding A3G. The energy required to stretch a polymer chain to a set tension, and the change in this value when the underlying polymer properties are

**Table 1.** Summary of key results and fitting parameters.

For FJC fitting parameters, stated plus/minus values are standard error of the mean based on fits to multiple FECs. For A3G conformation kinetics, stated plus/minus are error estimates for fitting parameters based on chi-squared minimization.

| FJC fits | Contour length (nm/nt) | Persistence length (nm) | Elastic modulus (pN) |
|---|---|---|---|
| ssDNA only | 0.565 ± 0.003 | 0.717 ± 0.024 | 804 ± 36 |
| A3G-ssDNA (HF) | 0.535 ± 0.006 | 1.40 ± 0.11 | 628 ± 70 |
| A3G Conformation Kinetics | Parameter | Model 1 | Model 2 |
| Compaction Rate | $k_C$(F = 0) (s$^{-1}$) | 2.8 ± 1.5 | 1.5 ± 0.8 |
| Compaction Distance | $d_C$ (nm) | 0.82 ± 0.21 | 0.88 ± 0.22 |
| Decompaction Rate | $k_{-C}$(F = 0) (s$^{-1}$) | 0.099 ± 0.017 | 0.099 ± 0.017 |
| Decompaction Distance | $d_{-C}$ (nm) | 0.077 ± 0.012 | 0.082 ± 0.013 |

altered as a constant force is maintained, can be calculated by integrating the difference in extension over force.

$$\Delta G(F) = -\int_0^F \left[ X_{final}(F) - X_{initial}(F) \right] dF \tag{9}$$

This ΔG gives a force-dependent energy barrier that the ssDNA-A3G complex must overcome to change its conformation. To define the extension profile of the ssDNA-A3G complex without A3G mediated compaction, $X_{-C}(F)$, we use the FJC fit to the high force FEC, where ssDNA is A3G-saturated but minimal compaction is observed. The main deviation from bare ssDNA is the increased persistence length exhibited by A3G saturated ssDNA, $p_{A3G}$ = 1.4 nm. The most likely cause of this reduction of ssDNA flexibility is the presence of large protein itself and we would expect this effect to persist to lower applied tensions. Indeed, the equilibrium FEC shows a decreased force plateau and increased extension at low forces consistent with increased persistence length. We next define the extension profile of the transition state, based on the FJC model (*Equation 1*), through which the ssDNA-A3G complex assumes the looped conformation by defining a minimum ssDNA contour length contraction, $d_C$, required before A3G assume its compacting conformation.

$$X_T(F) = \left( L_0 - \frac{d_C}{N} \right) \left( coth\left( \frac{2p_{A3G}F}{kT} \right) - \frac{kT}{2p_{A3G}F} \right) \left( 1 + \frac{F}{S} \right) \tag{10}$$

Similarly, there is a length of ssDNA extension, $d_{-C}$, that must occur for a compacted ssDNA-A3G complex to reach the transition state. Since $d_C$ and $d_{-C}$ are a length associated with each A3G monomer, these values must be normalized using the binding site size N to determine the average effect along the entire ssDNA strand. Using these calculated FECs (*Figure 4—figure supplement 1A*), we integrate over force to calculate ΔG, the change in the stretching energy of the ssDNA-A3G complex due to conformational change (*Figure 4—figure supplement 1B*). Finally, we use this energy cost to calculate the force-dependent rates of loop formation.

$$v_{obs}(F) = k_L(F) + k_{-L}(F) = k_L(0)e^{\frac{-\Delta G_L \cdot N}{kT}} + k_{-L}(0)e^{\frac{-\Delta G_{-L} \cdot N}{kT}} \tag{11}$$

This expression superficially resembles the Bell model, but the exponents now contain the integrated energy terms, which account for the polymer property dependent extension of ssDNA over all forces. We first plug into this updated model the same parameter set returned by the best fit of the Bell model (*Figure 4C*, red line), which shows a small but significant deviation for our measured values. However, by allowing the parameters to vary, we can reobtain a proper fit yielding a new parameter set. At zero force, A3G mediated compaction is greatly favored ($k_C$ = 1.5 ± 0.8 s$^{-1}$ compared to $k_{-C}$ = 0.099 ± 0.017 s$^{-1}$). Again, the transition length of A3G compaction (0.88 ± 0.21 nm) is much larger than the transition length of A3G decompaction (0.082 ± 0.012 nm).

This method of determining the free energy difference between FECs is also useful in examining the impact of binding site size on A3G binding kinetics. The change in extension observed during our constant force experiments involves both free A3G binding ssDNA and then equilibrating between its binding conformations depending on force. While the kinetics of this multistate process are complex, we can calculate the energy cost of A3G binding in its preferred conformation by integrating over the difference in extension between the FECs of A3G free ssDNA and ssDNA with equilibrated A3G (*Figure 1D*), yielding ΔG as a function of force (*Figure 4—figure supplement 1C*). This ΔG, which acts as an energy barrier to equilibrated A3G binding, can used to fit the rates observed during constant force incubation.

$$v_{obs}(F) = k(0)e^{\frac{-\Delta G_{eq} \cdot N}{kT}} \tag{12}$$

Since ΔG is calculated on a per nt basis but the kinetics of A3G binding and compaction are determined on a per protein monomer basis, we must define a binding site size N. The most direct way to experimentally measure binding site size is to observe whether a small DNA substrate of fixed size can accommodate the binding protein. Previously the binding and tetramerization of A3G to a 69 nt long ssDNA substrate has been measured using AFM imaging (*Pan et al., 2018*; *Shlyakhtenko et al., 2011*; *Shlyakhtenko et al., 2012*; *Shlyakhtenko et al., 2013*), suggesting a

minimal binding site size N ≤ 17 nt. In agreement, we find the kinetics of A3G binding are consistent with a binding site size on the order of 15 nt (*Figure 4—figure supplement 1D*). In contrast much smaller or larger values of N do not return the proper force dependence of observed A3G binding. At very low forces (≤10 pN), this rate dependence no longer holds as ssDNA self-interactions in the form of secondary structure and the formation of larger ssDNA loops may interfere with the idealized model of ssDNA acting as a simple array of sequential binding sites.

## Effects of ssDNA secondary structure on A3G binding

The above experiments are performed under conditions at which ssDNA is extended enough that the formation of secondary structure (e.g. nearby bases interacting to form local hairpins) are mostly excluded. Under physiological conditions, however, the absence of a strong extending force would allow such structures to form, including some ordered, sequence specific structures, such as the TAR hairpin. The fully ssDNA substrate used in the proceeding experiments cannot be used to simulate such conditions, as significant compaction of the ssDNA results in the labeled beads approaching one another and interfering with the highly flexible ssDNA. Instead, we use a hybrid dsDNA/ssDNA construct based on a design developed in the lab of Tom Perkins (*Okoniewski et al., 2017*). This construct consists of 6.5 kbp of dsDNA, but with one strand nicked at two specific locations one knt apart, such that after overstretching the construct for the first time, the strand segment between the nicks irreversibly dissociates into solution (*Figure 5A*). Thus, after 'force activation', the construct now consists of 1 knt of ssDNA flanked by dsDNA handles. The dsDNA handles, which are labeled and attached to the beads in the optical trapping system, are less flexible than the ssDNA and allow the ssDNA to fully compact (tension and extension approach zero), while maintaining a sufficient distance between the beads. As a result, the transition from the ssDNA being fully compacted (due to the formation of secondary structure) to fully extended (following the FJC model) is visible in a FEC (*Figure 5A*). This transition is observed in detail by adjusting the FEC's rate of extension change (*Figure 5—figure supplement 1A*). With faster stretching rates, the secondary structure releases at higher forces. In contrast, the return curves show less rate dependence as the ssDNA quickly compacts when given sufficient slack. Note, we perform all experiments using this construct in a buffer containing near physiological salt concentrations (150 mM Na and 1 mM Mg), which enhances the formation of ssDNA secondary structure. By averaging over multiple FEC curves, the degree of ssDNA compaction is calculated by interpolating the distance between the average FEC and the model lines for the hybrid construct with fully compacted and fully extended ssDNA (*Figure 5—figure supplement 1B*). This also determines the total change in ssDNA extension due to compaction (i.e. the reduction in extension exhibited by the average FEC versus what is predicted by the FJC model).

Next, we examine how A3G interacts with this measured ssDNA secondary structure. For control, we verify the binding of A3G does not impact the extension of dsDNA below the overstretching transition (*Figure 5—figure supplement 2*), so that all extension change observed for the hybrid construct can be attributed solely to the ssDNA region. As with the fully ssDNA construct, we hold the hybrid construct at a fixed force and incubate with 50 nM A3G (*Figure 5B*). At moderate forces (12 and 16 pN), where ssDNA secondary structure is mostly destabilized, we see a similar reduction in ssDNA length as in previous experiments. For low forces (4 and 8 pN), however, as A3G binds, the ssDNA extension increases as A3G physically interrupts secondary structure formation. The final extension, however, stops well short of that of fully extended ssDNA, indicating the formation of loops reduces the extension of the ssDNA-A3G complex. We again fit exponential functions to these extension over time curves to extract both a rate and amplitude of extension change (*Figure 5C and D*). Both the rate and amplitude values for the 12 and 16 pN curves agree with the 20 pN data obtained using the fully ssDNA construct. The lower force data exhibits larger still ssDNA contraction, similar to what is observed in the low force ssDNA-A3G FEC and requires a longer timescale to reach equilibrium.

As a final experiment, we allow A3G to bind ssDNA in the absence of tension (~0 pN). The presence of dsDNA handles allows the ssDNA region to be fully unextended, allowing for potentially more A3G mediated compaction, as opposed to the fully ssDNA substrate which always has nonzero extension. We cannot observe A3G binding in real time when the ssDNA is not extended but stretching the construct after the 100 s incubation period reveals structures formed by the ssDNA-A3G complex (*Figure 5E*). Even at moderate puling rates (100 nm/s), the A3G remains compacted

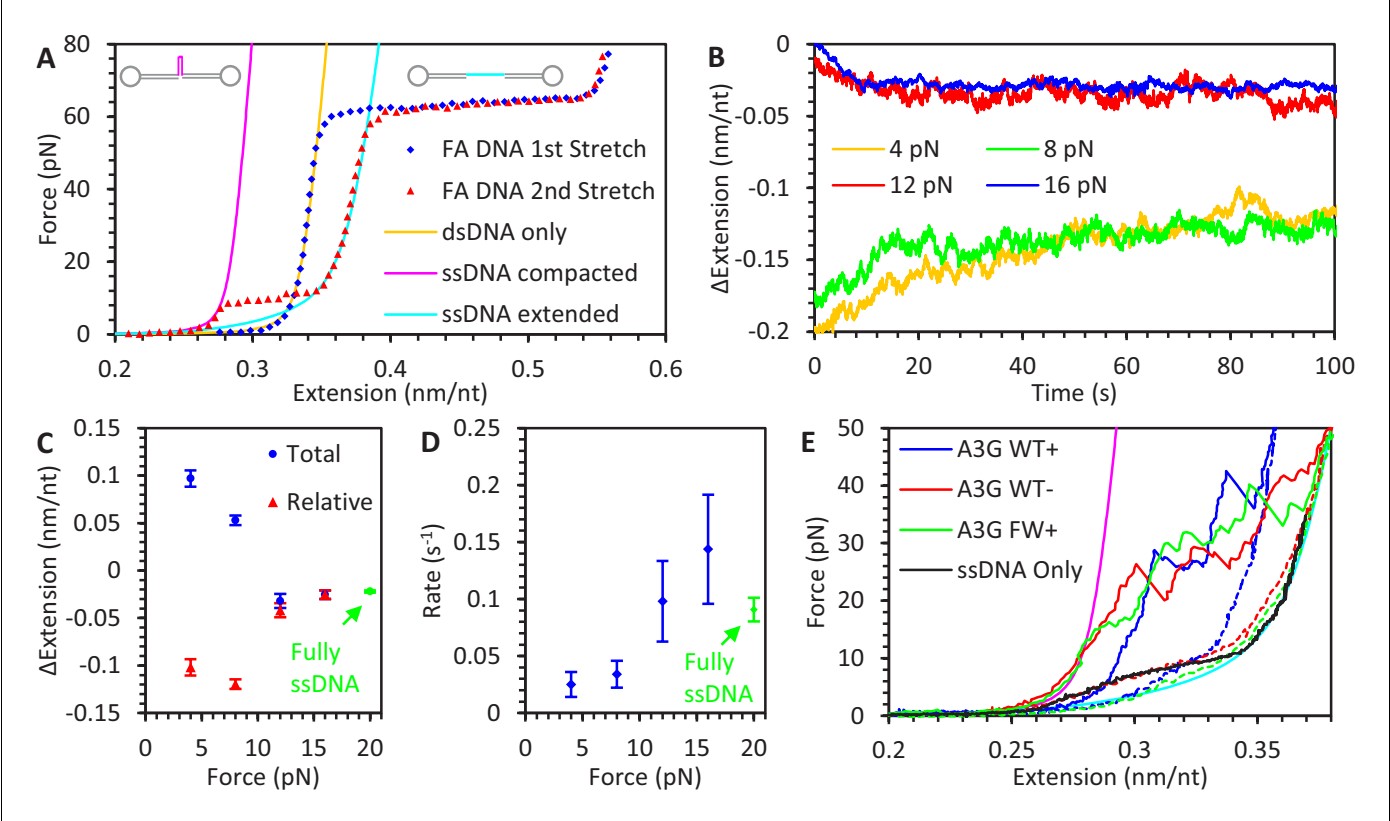

**Figure 5.** A3G binding to ssDNA/dsDNA hybrid construct. (**A**) FECs of hybrid construct. A 6.5 kbp dsDNA construct is stretched using optical tweezers (blue diamonds). The first stretch follows the WLC model (yellow line), but a one knt section of ssDNA between two nicking sites dissociates during overstretching, altering the FEC of subsequent stretches (red triangles). At low forces, secondary structure compacts the ssDNA and the constructs total extension is consistent with the ssDNA section having zero net extension (magenta line). At higher forces, secondary structure is disrupted, and the ssDNA's contribution to the construct's total extension follows the FJC model (cyan line). (**B**) ssDNA extension change due to A3G binding at low force. The hybrid construct is held at a constant force and incubated with 50 nM A3G. At the lowest forces (4 pN in yellow and 8 pN in green), ssDNA is initially compacted due to secondary structure but increases it extension as A3G binds. (**C**) Average ssDNA extension changes. The average total extension changes (compared to initial extension, blue circles) and average relative extension changes (compared to FJC model, red triangles) are plotted as a function of force. Above ~10 pN, ssDNA extension is decreased similar to experiments using the fully ssDNA construct at 20 pN (green symbols). At lower forces ssDNA extension appears to increase, but when the initial decrease in extension due to ssDNA secondary structure prior to A3G incubation is taken into account, the final ssDNA-complex is still shorter than bare, linear ssDNA. (**D**) Average rate of extension change. While the timescale required for the ssDNA to reach an equilibrium length at forces above ~10 pN agrees with 20 pN data obtained using a fully ssDNA construct, this process is greatly slowed at low forces where significant ssDNA secondary structure is present. (**E**) Stabilization of ssDNA loops by A3G. After incubating the hybrid construct with A3G at near zero force so that the ssDNA region is completely unextended, the DNA is stretched at a rate of 100 nm/s. As compared to the compaction of ssDNA due to intrinsic secondary structure formation (panel B), much higher forces are required to extend the ssDNA once A3G is bound (blue line). This effect remains even after removing free A3G from solution (red line). Similar effects are seen when incubating with FW mutant A3G (green line), indicating this A3G looping does not require A3G oligomerization. Error bars are standard error based on multiple experimental replications (N ≥ 5) with different ssDNA molecules.

The online version of this article includes the following source data and figure supplement(s) for figure 5:

**Source data 1.** Numerical values and experimental replicates for data plotted in **Figure 5**.
**Figure supplement 1.** Kinetics and equilibrium of ssDNA secondary structure formation.
**Figure supplement 2.** A3G binding to dsDNA.

to higher forces before sudden drops in force and subsequent increases in extension are observed at forces between 20 and 50 pN, consistent with large ssDNA loops, stabilized by multiple A3G monomers binding in parallel, suddenly being pulled apart. These curves also appear more jagged than the stretch curves of the fully ssDNA substrate (**Figure 1C**) because the A3G-ssDNA complex is initially equilibrated at near zero force allowing further compaction, the ssDNA is 8X shorter (resulting in less averaging over length), and the stiff dsDNA handles allow for a greater drop in force after

an extension increase event. Furthermore, even after rinsing the DNA in protein free buffer for 200 s, these loops are still observed. We again repeat these experiments using the oligomerization-deficient FW mutant A3G, which also forms similar structures, confirming that this looped binding conformation is not dependent on A3G oligomerization. The exact shape of these FECs are highly variable, even when stretching the same DNA template multiple times, suggesting that the A3G in binding in a disordered manner. But even cursory examination of the FECs reveals singular events involving sudden increases in extension greatly in excess of the ~1 nm loop size derived in the force jump experiments, where forces of >10 pN are maintained. This suggests A3G is forming much larger ssDNA loops (large enough to accommodate multiple A3G monomers judging by the loop stability) in the absence of DNA tension, similar to conditions during viral replication.

## Discussion

### The two conformations of the ssDNA-A3G complex

Our results show that A3G can bind ssDNA in both a ssDNA contracting state and a non-contracting state. Furthermore, these states are well represented as two discrete states separated by a single energy barrier of ~3 $k_b$T (*Figure 4*). The exact physical mechanism behind these interactions, however, requires further interpretation. There is currently no resolved structure of full-length A3G in complex with nucleic acids and our assay monitors only the extension of the ssDNA substrate, not the state of the protein itself. Nevertheless, we can produce a simple model of the A3G-ssDNA interaction that is consistent with both these new findings and previous experimental evidence. The ability of A3G to greatly contract an ssDNA substrate under low applied tension is likely a result of its two-domain structure. While the CTD is enzymatically active, it has very low binding affinity for ssDNA when isolated from the NTD (*Chelico et al., 2010*). In fact, the A3G CTD is most structurally similar to the single domain A3A, which is also enzymatically active despite its low ssDNA binding affinity (*Love et al., 2012*; *Shandilya et al., 2014*). When attached to the non-enzymatic NTD domain, however, the entire protein has high binding affinity, which we measure here between 1 and 10 nM. Thus, while the CTD is required for catalytic activity, the NTD is primarily responsible for stable binding to ssDNA. With both domains having different yet vital functions, taking into account both their interactions with nucleic acid substrates is necessary to describe A3G function. Correspondingly, rather than describing the entire A3G monomer as ssDNA bound, we must distinguish which of the two domains are actually binding the ssDNA substrate. If both domains are bound simultaneously, this will place additional constraints on the ssDNA conformation. This could result in the ssDNA extension increasing if longer segments of the substrate are forced into a straight form (such as the segment between the two domain binding sites) or decreasing if forcing the ssDNA to conform to multiple binding surfaces decreases its effective contour length. Additionally, an even larger reduction in extension could be seen if the two domains bind and bring together non-neighboring ssDNA sites, forming an ssDNA loop. When the ssDNA substrate is saturated with protein or when straightened by force, available binding sites for the CTD are limited to directly adjacent to NTD, resulting in small loops on the order of 1 nm. However, in the absence of applied tension, the flexibility of ssDNA allows the two domains to bind two different sites along the viral genome that are not directly adjacent, forming large looped structures (*Figure 5E*).

Additional evidence of A3G's ability to greatly contract ssDNA substrates is provided by other biophysical experiments. AFM measurements of A3G-ssDNA bonds disrupted by forces on the order of 50 pN observed two distinct populations of rupture events (*Shlyakhtenko et al., 2015*). A short rupture equivalent to the release of ~18 nt was seen under variety of conditions, consistent with the compact binding site size required for an A3G tetramer to form on a 69 nt segment of ssDNA (*Shlyakhtenko et al., 2011*). When a longer ssDNA substrate was used, however, larger rupture events, consistent with the release of longer ssDNA segments, were observed. This mode of binding was further enhanced by the presence of the target CCC deamination motif in the substrate. Similar results were observed in FRET experiments observing interactions between an A3G monomer with a labeled CTD and an ssDNA substrate with a label near its 3′ or 5′ end (*Senavirathne et al., 2012*). A wide range of observed FRET efficiencies indicated that A3G was highly mobile on the substrate while scanning. Additionally, A3G was able to highly contract a 72 nt long substrate in the presence of a CCC motif, which was deaminated by the A3G. Taken together, these results suggest that A3G

is able to bind and contract a large segment of ssDNA in excess of A3G's minimum binding site size and that this binding conformation is associated with enzymatic activity. Thus, we conclude that A3G's highly contracting binding conformation is associated with an enzymatically active state in which the CTD is fully engaged with the ssDNA.

An A3G-ssDNA complex assuming its preferred structure (with the CTD engaged) may be primarily dependent on the flexible substrate conforming to the structure of the protein itself. Conformational change of the A3G monomer itself, however, may also aid in this binding process. Molecular dynamics simulations of an isolated A3G monomer have predicted the existence of a 'globular' state, with the two domains in close contact, and a 'dumbbell' state, with a seven amino acid linker between the two domains separated by up to 1 nm (*Gorle et al., 2017*). AFM imaging is able to independently resolve the centers of the two domains for A3G in the dumbbell conformation but not for A3G in the globular conformation. Both simulation and AFM results show that the globular state is favored (~90%) in the absence of a nucleic acid substrate. AFM imaging has shown, however, that the presence of an ssDNA substrate greatly stabilizes the dumbbell binding conformation (*Pan et al., 2019*). On a 69 nt long ssDNA substrate, an A3G monomer is more likely to assume the dumbbell conformation. When the substrate is shortened to 25 nt, however, the globular state is more prevalent (though not to the same degree as in the complete absence of nucleic acid). Additionally, the protein is observed occasionally with only one domain in contact with the ssDNA, with the other domain free and transitioning between the dumbbell and globular states on the timescale of 1 s. These results are consistent with our presented model in which the NTD is stably bound to the ssDNA for extended periods of time while the CTD transiently samples and binds the available substrate. These results also suggest that free A3G, which is primarily in a globular conformation, first binds ssDNA at a single site. Once bound, however, the dumbbell conformation is stabilized, as this allows the CTD to bind independent of the NTD. However, the CTD cannot bind stably if the NTD is positioned too close to the end of the substrate, similar to our observations that excessive straitening of the ssDNA substrate prevents contractive binding of A3G.

## Kinetics of A3G binding and deamination

Our experiments measure a wide variety of rates governing the A3G-ssDNA interaction. Extrapolated to zero-force, we find that A3G initially binds ssDNA at a concentration-dependent rate of $0.01 \text{ s}^{-1} \text{ nM}^{-1}$ and dissociates as a monomer on a timescale of ~70 s. The A3G-ssDNA complex switches between its two conformational states on timescale of ~1–10 s, allowing for many transitions during a single binding event. This is in good agreement with FRET experiments, which measured two timescales of interaction between ssDNA and 1 nM A3G (*Senavirathne et al., 2012*). On the short timescale, on and off rates were observed at $0.01 \text{ s}^{-1}$ and $0.23 \text{ s}^{-1}$ respectively. This on rate matches our measured ssDNA initial binding rate. The short timescale off rate is likely the result of the transient binding of the CTD, which in turn controls the rate at which A3G can sample the two binding conformations. Long FRET trajectories lasted an average of 75 to 110 s, depending on the exact ssDNA substrate, consistent with our 70 s dissociation timescale, which is governed by the stable binding of the NTD. Thus, while the NTD remains bound to the DNA substrate, the CTD is able to rebind dozens of times as it scans for deamination targets. At reduced salt concentrations, which reduces electrostatic screening between the NTD and the ssDNA substrate, a smaller range of FRET states were observed per binding event. Presumably this slows A3G's scanning ability via sliding, as the NTD remains stationary, preventing the CTD from sampling the entire substrate. This is consistent with our results showing weaker initial, NTD mediated binding (as evidenced by a slower on rate, *Figure 2—figure supplement 1*) at higher salt concentrations. Interestingly, salt concentration also modulates interprotein interactions, with AFM imaging of 100 nM A3G alone observing more oligomers in the absence of 5 mM Mg, but when incubated with 200 nM 70 nt long ssDNA, more oligomers are observed in the presence of 5 mM Mg (*Chelico et al., 2008*). However, we do observe WT A3G stably binding ssDNA over a range of salt concentrations (*Figure 2—figure supplement 1*), suggesting oligomerization is a robust process, albeit with slightly altered kinetics.

## Impacts of A3G oligomerization

Since A3G tends to form oligomers while bound to an ssDNA substrate, it is important to understand how oligomeric state impacts protein function. Under the conditions used in our experiments

(a single, long ssDNA substrate exposed to a low concentration but effectively inexhaustible supply of free A3G), A3G oligomerization drastically increases the time over which A3G remains bound to the substrate. However, applying high force (>30 pN) to the ssDNA substrate allows WT A3G to dissociate at rates equal to that of the oligomerization deficient FW mutant (*Figure 2F*). Applied tension destabilizes oligomerization, possibly by preventing neighboring proteins from properly aligning key oligomerization interfaces when the ssDNA substrate is straightened. At moderate forces, both WT A3G and FW mutant A3G contract the ssDNA substrate to similar degrees. As applied force approaches zero, however, the A3G is able to create larger ssDNA loops at a faster rate in the absence of oligomerization (*Figure 3*), suggesting inter-protein interactions slow down the CTD's ability to repeatedly bind the substrate.

How A3G's oligomerization impacts its enzymatic activity has been a particular matter of debate. Typically, the rate of an enzymatic reaction will increase linearly along with the concentration of the enzyme itself. For A3G, however, increased concentration also results in increased oligomerization, leading to conclusions that oligomerization does not inhibit enzymatic activity (*McDougall et al., 2011*; *Pan et al., 2018*; *Pan et al. (2018)* used AFM to measure A3G oligomerization via protein complex volume on a 70 nt ssDNA substrate, while in parallel measuring the deamination of a single target site on a similar ssDNA construct under the same conditions at the same A3G concentrations. They observed that both A3G oligomerization and deaminase activity increased with A3G concentration, though the increase in effective deamination rate was much smaller than the increase in A3G concentration. The study claims that each oligomeric state of A3G exhibits similar enzymatic activity as an A3G monomer (*Pan et al., 2018*). A3G's function in vivo occurs under very different conditions, as each packaged A3G monomer must locate multiple target sites of deamination along the nearly 10 kbp viral genome. If this trend persisted, however, A3G oligomerization would slow enzymatic activity as the number of A3G monomers packaged in the virion is strictly limited such that the formation of oligomers reduces the total number of A3G complexes (e.g. 8 A3G monomers would deaminate the viral DNA faster than the two tetramers that could be formed by oligomerizing the same number of monomers).

Another difficulty in simultaneously measuring both oligomerization and deaminase activity is that they may occur on different timescales. For instance, A3G could either deaminate a substrate as monomers and then oligomerize or oligomerize on the substrate and then deaminate it. The former model is supported by experiments where A3G is preincubated with unlabeled ssDNA to allow for oligomerization before the addition of the ssDNA substrate containing the deamination target. The pre-oligomerized WT A3G has greatly reduced enzymatic activity, while the non-oligomerizing FW mutant A3G and non-oligomerizing A3A retain a faster enzymatic rate (*Adolph et al., 2017*; *Morse et al., 2017*). This interplay between deamination and oligomerization can also be seen in high concentrations of A3G inhibiting its own deaminase activity. When a saturating concentration of 500 nM A3G is incubated with an ssDNA substrate with a single CCC motif 36 nt from the 3′ end, half the target sites are deaminated on the 10 s timescale (*Chelico et al., 2010*). The remaining sites are slowly deaminated over the course of 100 s of seconds. In contrast, the FW mutant A3G continues to deaminate these remaining sites with a much smaller reduction in speed. Here, while the high concentration of A3G initially allows for the rapid deamination of the substrate, over time oligomers are formed, slowing subsequent reactions. Experiments with force-melted dsDNA demonstrate that interstrand binding causes A3G to switch from an oligomerized state to interstrand bound state also on the timescale of 100 s of seconds. The exact timescale of this effect likely varies due to the number of ssDNA-ssDNA contacts (e.g. two strands of ssDNA held closely in parallel, a single long strand of flexible ssDNA folding onto itself, or many short ssDNA oligos colliding in solution). All of these conditions are consistent with a model in which, over a long timescale, A3G oligomers can be broken up due to intersegmental binding, which allows A3G to proceed with enzymatic activity, albeit at a reduced rate. It should be noted that in our experiments, which contain an effectively infinite supply of free protein relative to available ssDNA, the ssDNA is fully saturated such that monomers that break off from bound oligomers are to likely reform with other bound proteins before fully dissociating. In contrast, in AFM experiments where the ratio of A3G monomers to substrates are limited and short ssDNA binding substrates prevent the formation of excessively large oligomers, the breaking up of oligomers can be directly observed (*Shlyakhtenko et al., 2013*). Thus, we would expect the formation and disbanding of oligomers under physiological conditions to be dynamic. Additionally, the small number of A3G monomers packaged in the virion, in comparison to the long

viral DNA genome, will decrease the rate of collisions between A3G monomers and their resulting oligomerization. This would allow A3G to remain monomeric for longer times than we observe in our experiments, such that the majority of the viral genome can be sampled prior to the formation of stable oligomers.

## Biophysical model of A3G function

By synthesizing these new results with those of previous studies on A3G binding and enzymatic activity, we propose a detailed model of A3G function (*Figure 6*). Due to the small size of the virion, the effective concentration of A3G is very high and once the viral RNA is degraded, the (-) ssDNA is immediately bound by free A3G, which is predominately in a globular conformation. The initial binding state is driven by the affinity of the NTD and does not require the CTD to bind specifically. On the timescale of 1 s, the CTD binds the ssDNA substrate, enabled by partial decoupling of the two domains when the monomer assumes a dumbbell conformation. With the CTD bound, A3G assumes an enzymatically active conformation, allowing for the specific recognition and deamination of CCC target motifs. The A3G monomer continues scanning for more potential deamination sites, as the NTD slides along the ssDNA substrate while the CTD repeatedly releases and reengages the ssDNA substrate. Since the NTD remains bound on the timescale of 100 s, an A3G monomer is able to deaminate multiple sites in close proximity during a single binding event, leading to processive deamination. A3G exhibits strong directional bias in its deamination activity, with enzymatic activity requiring the CTD to bind towards the 5′ end of minus-strand ssDNA relative to the NTD. This

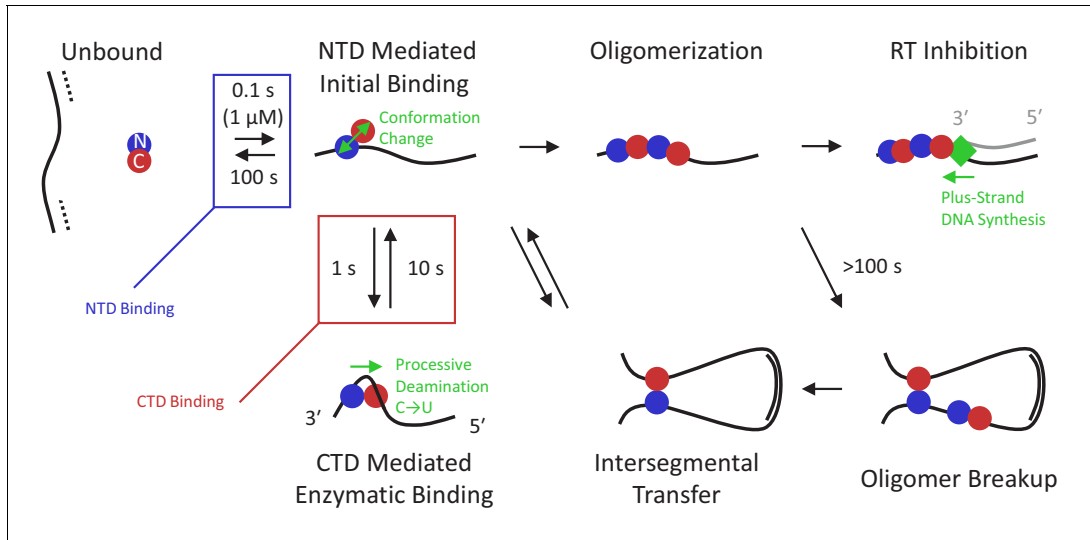

**Figure 6.** Multi-state model of A3G enzymatic activity. Once the viral RNA template has been degraded (dashed line), the minus-strand ssDNA (solid line) is available for A3G binding. Due to the high effective concentration of the few A3G molecules packaged in the small volume of the virion and the strong electrostatic interaction between the NTD and negatively charged ssDNA, free A3G (predominately in a globular conformation) quickly binds available ssDNA and remains bound on the timescale of 100 s. While bound, A3G monomers can spontaneously switch into a dumbbell conformation with the two domains partially decoupled, enabling transitions along three paths. First, the CTD can locally bind the ssDNA in an enzymatically active state, enabling recognition of target CCC motifs and subsequent deamination activity. This binding conformation stabilizes A3G's dumbbell form and tightly contracts the ssDNA substrate. Quick sampling of the ssDNA by the CTD, along with the ability of the NTD to slide along the ssDNA under physiological salt conditions, allows fast and processive deamination of neighboring target sites before full A3G dissociation. A3G deamination activity displays a strong directional bias, inhibited from accessing sites within ~30 nt of the 3′ end due to the presence of the NTD, and deaminating processively in the 5′ direction. Second, the CTD can bind non-locally to another region of the viral genome due to the high flexibility of ssDNA segments. Due to the stronger ssDNA binding affinity of the NTD versus the CTD, the protein usually remains at the NTD side of the loop, but sometimes relocates when the CTD remains bound longer. This allows for moderately processive deamination even in the presence of barriers such as regions of dsDNA and RNA/DNA hybrid. Third, if an A3G monomer collides with other bound protein(s), an oligomer will form. This path becomes favored at higher A3G concentrations and over longer timescales. Oligomerization reduces deaminase activity due to decreased protein mobility and/ or occlusion of the CTD's active site by other bound A3G subunits. Oligomers can be broken up, however, over long timescales, allowing for additional enzymatic activity at a reduced rate. Alternatively, oligomers that remain bound to the minus-strand ssDNA template block processive polymerization of the complementary plus strand ssDNA (gray line) by RT (green diamond). Slowing RT also extends the window of ssDNA vulnerability, allowing other A3G monomers to fully deaminate the viral genome.

inhibits deamination of sites within ~30 nt of the 3′ end of an ssDNA segment (where the NTD occludes CTD binding) and biases processive deamination towards the 5′ direction. When the long flexible viral DNA folds on itself, randomly forming DNA-DNA contacts, the CTD can also bind non-locally, forming large DNA loops. This binding state is only transient, as once one of the domains releases the substrate, the protein will be localized at the site of the domain that remains bound. Due to the high affinity of the NTD, the protein is most likely to remain at its original position. This allows A3G to processively deaminate nearby sites even in the presence of excess free DNA substrate (*Adolph et al., 2017*). However, on the occasions where the NTD releases the substrate before the CTD, the protein will undergo intersegmental transfer, allowing A3G to directly move from one region of ssDNA to another, without fully dissociating and diffusing into bulk solution. As the viral DNA exists as a combination of ssDNA, dsDNA, and RNA/DNA hybrid during proviral DNA synthesis, a mechanism to bypass non-single-stranded regions is critical for effective deamination. Due to the low number of A3G monomers packaged in the virion, with only one monomer per ~1 kilobase of the viral genome (*Xu et al., 2007*), collisions between two bound proteins are rare. However, when two monomers eventually do come into contact on the ssDNA substrate at longer time-scales, a dimer is readily formed. Oligomerization inhibits further enzymatic activity by slowing the mobility of the protein complex. Over the course of 100 s of seconds, however, the oligomer will eventually breakup as the competitive binding at ssDNA-ssDNA contacts promotes the intersegmental binding conformation. Otherwise, the formed oligomers remain stably bound, slowing reverse transcription at that site. The stalling of RT in turn allows the remaining enzymatic A3G to continue deamination of exposed ssDNA, ensuring maximal damage to viral genome integrity.

## Materials and methods

### DNA binding templates

For the fully ssDNA construct, an 8.1 kbp dsDNA construct with digoxigenin (DIG) and biotin labeled ends with a free 3′ end was constructed as previously described (*Figure 1A*) (*Naufer et al., 2017*). The dsDNA vector pBacgus11 (gift from Borja Ibarra) was linearized using high fidelity restriction enzymes BamHI and SacI (New England Biolabs). A dsDNA handle with digoxigenin (DIG) labeled bases with a complementary end to the BamHI sequence was PCR amplified (*Ibarra et al., 2009*). The DIG handle and a biotinylated oligonucleotide with a 3′ end complementary to the SacI sequence (Integrated DNA Technologies) were ligated to the linearized pBacgus11 using T4 ligase (NEB). For the dsDNA/ssDNA hybrid construct, a 6.5 kbp fragment contained in a plasmid (gift from Tom Perkins) containing a designed sequence (*Okoniewski et al., 2017*) was PCR amplified using KOD Hot Start DNA Polymerase (Novagen). One end of product contains a biotinylated primer, and the other end was cut and ligated to same DIG handle described above. The labeled construct was then nicked using restriction enzymes Nt.BspQI and Nb.BsmI (New England Biolabs).

### A3G purification and preparation

A3G protein was expressed in baculovirus and constructed by PCR amplification of the coding portion of A3G from IMAGE clone 4877863 (ATCC) (*Chelico et al., 2010*). Recombinant baculovirus production for expression of glutathione S-transferase (GST)-A3G in Sf9 cells was carried out using the transfer vector pAcG2T (BD Biosciences). Sf9 cells were infected with recombinant virus at a multiplicity of infection (MOI) of 1 for 72 hr. A3G was purified in the presence of RNase A and the GST tag cleaved on the affinity column (*Chelico et al., 2010*). The A3G is approximately 95% pure by SDS-PAGE.

### Optical tweezers measurements of A3G binding

The 8.1 kbp dsDNA construct was tethered between a 2 µm anti-digoxigenin coated bead and a 3 µm streptavidin-coated bead (Spherotech) held in place by a micropipette tip and a dual beam optical trap, respectively. The micropipette tip was moved by a piezo electric stage with 0.1 nm precision to change the extended length of the DNA while the deflection of the laser trap was measured to calculate the force exerted on the trapped bead and thus the tension along the DNA. Additionally, a bright-field image of the two beads was recorded at 40X magnification. The instrument is controlled via a National Instruments data acquisition card and a custom written program in Lab

Windows. In order to create an ssDNA binding template, T7 DNA polymerase was flowed into the sample and the DNA was held at a constant force of 50 pN to trigger exonucleolysis (*Wuite et al., 2000*). To measure A3G binding to the ssDNA, the template was first held at a constant force between 20 and 80 pN in a buffer containing 10 mM HEPES and 50 mM $Na^+$ at pH 7.5, except where specifically noted. A3G diluted to a concentration of 50 nM in the same buffer was flowed into the sample and as constant force was maintained, the ssDNA's extended length was shortened as A3G binds (*Figure 1C*). After 100 s incubation time, the A3G was displaced with clean buffer to allow for dissociation of loosely bound protein, resulting in a partial return to the original bare ssDNA length. All experiments were repeated multiple times as a distinct biological replication (using a new DNA substrate and A3G dilution) to establish a mean value and standard error, represented by the error bars in plotted figures. The exact number of replicates for each condition is included in supplied source data files.

## Analysis of DNA extension data

To correct for drifts in the system, which can alter the position of the micropipette tip as elements of the instrument expand and contract due to slight thermal fluctuations, the instantaneous position of the two beads tethered to the DNA construct were determined using acquired using brightfield images. The distance between the beads' centers were measured in each frame, and he change in this distance over time was compared to the movement of the piezo electric stage controlling the micro-pipette tip to determine time-dependent drift (in the absence of drift, both sets of values are identical). This drift correction was then subtracted from the extension data produced by the piezo electric stage. The final DNA extension values retain both the 0.1 nm resolution provided by the piezo and the >1000 s timescale stability in extension values determined by image analysis. All acquired data was analyzed using custom Matlab code to extract the simultaneous force and extension of the DNA construct.

## Acknowledgements

We thank Tom Perkins (University of Colorado) for the gift of the plasmid used to create the dsDNA/ssDNA hybrid construct and Penny Beuning, Bilyana Koleva, and James McIsaac (Northeastern University) for amplifying the plasmid and technical advice in constructing the construct.

## Additional information

### Funding

| Funder | Grant reference number | Author |
|---|---|---|
| National Institute of General Medical Sciences | GM072462 | Michael Morse Ioulia Rouzina Mark C Williams |
| Canadian Institutes of Health Research | MOP137090 | Yuqing Feng Linda Chelico |

The funders had no role in study design, data collection and interpretation, or the decision to submit the work for publication.

### Author contributions

Michael Morse, Conceptualization, Software, Formal analysis, Investigation, Visualization, Methodology; M Nabuan Naufer, Resources, Software; Yuqing Feng, Resources; Linda Chelico, Resources, Funding acquisition; Ioulia Rouzina, Methodology; Mark C Williams, Conceptualization, Supervision, Funding acquisition, Project administration

### Author ORCIDs

Michael Morse https://orcid.org/0000-0002-8561-1833
Mark C Williams https://orcid.org/0000-0003-3219-376X

Decision letter and Author response
Decision letter https://doi.org/10.7554/eLife.52649.sa1
Author response https://doi.org/10.7554/eLife.52649.sa2

## Additional files

**Supplementary files**

• Source code 1. UTC – Universal Tweezer Controls: LabWindows/CVI program used to control and collect data from optical tweezers system.

• Source code 2. Fexta – Force Extension Analysis: Matlab program used to analyze force and extension data collected by optical tweezers system.

• Source code 3. imFexta – Image Analysis: Matlab program to analyze bright field images of trapped beads to correct for long term thermal drift in optical tweezers system.

• Transparent reporting form

**Data availability**

Source data files have been supplied for Figures 2, 3, 4, 5 and Figure 2—figure supplement 1. Additionally, custom written Matlab and Lab windows code is supplied with this manuscript.

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
