## [Decision Letter]

**Acceptance summary:**

Understanding how Apobec3G (A3G) interacts with DNA may provide important clues to resolve the mystery of how it manages to efficiently deaminate the HIV genome. Here, Morse et al. used single-molecule force spectroscopy (optical tweezers) to measure the binding and unbinding kinetics of the restriction factor APOBEC3 (A3G) under a broad range of mechanical tensions. The authors constructed a detailed molecular model of the binding and diffusion of individual molecules and oligomers of A3G on ssDNA. One interesting feature of the proposed model is that A3G binds single-stranded DNA (ssDNA) in multiple steps and in two distinct conformations. The enzymatically "active" state of A3G corresponds to the contracted conformation of A3G-DNA complex where the catalytic CTD is fully engaging the DNA. Overall, the reviewers agree that this this will be a valuable *eLife* paper especially given the care and rigor with which they've handled investigating the protein binding modes coupled to the polymer properties.

**Decision letter after peer review:**

Thank you for submitting your work entitled "HIV restriction factor APOBEC3G binds in multiple steps and conformations to search and deaminate single-stranded DNA" for consideration by *eLife*. Your article has been reviewed by three peer reviewers, one of whom is a member of our Board of Reviewing Editors, and the evaluation has been overseen by and a Senior Editor. The reviewers have opted to remain anonymous.

Our decision has been reached after consultation between the reviewers. Based on these discussions and the individual reviews below, we regret to inform you that your work will not be considered further for publication in *eLife*. However, if your model holds after addition of necessary experiments and re-evaluation of the data along the lines suggested by the reviewers, or if a new model emerges, the reviewers and I would encourage you to resubmit the paper.

Summary:

Understanding how Apobec3G (A3G) interacts with DNA may provide important clues to resolve the mystery of how it manages to efficiently delaminate HIV genome. Morse M. et al. use single-molecule force spectroscopy (optical tweezers) to measure the binding and unbinding kinetics of the restriction factor APOBEC3 (A3G) under increasing mechanical tensions. The authors seek to construct a detailed molecular model of the binding and diffusion of individual molecules and oligomers of A3G on ssDNA. One of the major claims is that A3G binds single-stranded DNA (ssDNA) in multiple steps and in two distinct conformations. The reviewers agreed that overall, the paper is well written and data seems consistent. However, the theoretical model and the major conclusions of the paper are based on indirect measurements, experiments performed at moderate or high tensions on ssDNA (10-80 pN) and protein concentrations favoring oligomerization. This calls into question whether the results could be extrapolated to physiological conditions. The authors do not measure individual A3G molecules or complexes, but the mean activity of very many of them on a single ssDNA tether. They also do not directly measure protein motion or diffusion. The reviewers also have concerns with how the authors consider polymer properties of the protein-bound DNA.

Because of these concerns, the reviewers fell that in its current state the manuscript does not meet the standard for *eLife* publication.

The concerns of the reviewers are detailed in the individual critiques below. Here is the summary of the additional work that is needed to strengthen the study:

1) Additional work at forces that allow secondary structure formation (F< 6pN) may help to understand better the binding properties of A3G on ssDNA. Since the model proposed by the authors made some predictions for the protein-DNA organization at low forces, it would be great if they could corroborate those.

2) Also, the theoretical model should be reconsidered; the revised model should take into account all suggestions provided by the reviewers. Among others, the critical points to consider would be:

• the values of the persistence length and stretch modulus parameters

• the protein binding site size, and its force dependence

• the effect of ssDNA secondary structure on protein binding

• amount of ssDNA covered by A3G at different forces and concentrations

• Full A3G-DNA force-extension curves should be provided, together with the corresponding fits with the model

• re-evaluation of the distance to the transition state with Dudko et al. (2008) models

Reviewer #1:

Understanding how Apobec3G (A3G) interacts with DNA may provide important clues to resolve the mystery of how it manages to efficiently delaminate HIV genome. In this work, the authors applied optical tweezers single-molecule technique to interrogate the A3G binding to ssDNA tethered between two beads. This follows the authors previous study that used DNA melted by application of force (2x DNA). Here, the ssDNA was produced by the T7-endonuclease activity, which resulted in a single ssDNA molecule between the two beads and allowed the authors to carry out binding experiments at broader range of forces and to build a detailed energy landscape of A3G-DNA interactions in two different conformational states. Additionally, comparing the results of 1x and 2x DNA the authors were able to conclude that A3G can also bridge the two DNA molecules, or at low or zero forces to stabilize the ssDNA loops.

Overall, the experiments are expertly performed, the authors' conclusions are interesting and are in agreement with previously published data by the authors and others (in particular with the FRET and AFM data). The model of two binding modes and the conclusion that the enzymatically "active" state of A3G corresponds to the contracted conformation of A3G-DNA complex where the catalytic CTD is fully engaged is reasonable.

Major issues that need to be addressed:

1) The authors' argument that physical effect of A3G binding must be a reduction in the DNA counter length is plausible and quite likely. One cannot dismiss, however, another possibility, namely that the change in the elastic stretching modulus upon A3G binding may also play a role.

2) Quantitative description of the A3G binding energy landscape depends on the A3G DNA binding site, which the authors estimate to be 18 nt based on the overall size of the protein. This estimation needs an experimental confirmation.

Reviewer #2:

Morse M. et al. use single-molecule force spectroscopy (optical tweezers) to measure the binding and unbinding kinetics of the restriction factor APOBEC3 (A3G) under increasing mechanical tensions. One of the major claims is that A3G binds single-stranded DNA (ssDNA) in multiple steps and in two distinct conformations. Overall, the paper is well written and data seems consistent. However, the theoretical model and the major conclusions of the paper are based on experiments performed at moderate or high tensions on ssDNA (10-80 pN) and protein concentrations favoring oligomerization, which calls into question whether the results could be extrapolated to physiological conditions.

Main points:

1) It is not clear why the authors did not perform experiments at tensions F<10 pN (which are more physiologically relevant). One of our major concerns is that at low tensions (<10 pN), the overall structure (organization) of ssDNA is governed by its large entropic elasticity and its propensity to form secondary structure: secondary structure may locally inhibit/control A3G binding to ssDNA (i.e. making it difficult to form oligomers) and entropic elasticity may additionally modulate binding modes and kinetics. Although the theoretical model includes the effects of entropic elasticity of the ssDNA through the FJC model, the putative effects of secondary structure on protein binding were not experimentally tested nor modeled. The role of the secondary structure of ssDNA would be even more relevant in experimental conditions with higher salt concentrations because increasing concentrations of Na^+^ and Mg^2+^ are known to strongly stabilize the secondary structure of nucleic acids. The authors observed that binding and oligomerization is robust under high ionic strength conditions (subsection “Force dependence of A3G binding kinetics”, Figure 2—figure supplement 1) and claimed that the behaviors observed in their study should be observed under physiological conditions and in vitro biochemical assays. This is arguable because they performed all the experiments at tensions that prevent the formation of secondary structure, assuming no role of this phenomenon on protein binging/unbinding kinetics. Therefore, extrapolation of their model to zero force conditions (physiologically relevant) may not be entirely correct. The authors should comment on why did not conduct experiments at low forces and what the effect of secondary maybe on results and conclusions.

2) The theoretical model assumes some points that should be reconsidered:

A) The authors assume full protein coverage of the ssDNA at all tensions. However, at high forces the ratio between the measured Kd(f) and the experimental protein concentration, c, [Kd(f) / c = 10nM/50 nM = 0.2], indicates that at least 20% of the ssDNA chain would not be covered. This, in turn, would imply errors of this order or higher for the reported parameters. Could the 'putative' differences in protein coverage reduce the differences measured in the extension change per bound protein? (Less coverage would lead to less change in extension). The authors should consider including in their model protein coverage at its force dependence.

Protein coverage could be determined by the comparison of the binding energy with the elastic energy difference between the covered and uncovered segment ΔG(elastic, covered)-ΔG(elastic, uncovered). Elastic energy at constant force would be given by ΔGelastic=∫x(f)df, with x(f) the force-extension curve for each of the two states (covered and uncovered).

B) Equation 7 assumes that the persistence length of the A3G-ssDNA polymer equals the persistence length of naked-ssDNA. However, this may not be the case; the persistence length may be of the order of the size of the A3G monomer (or higher, for cooperative binding).

3) Equation 7 is supposed to fit the force-extension curve (FEC) of the A3G-ssDNA polymer. However, the authors do not show the final FECs of the protein-DNA complexes nor their corresponding fits with the model. Fits to the FECs of the A3G-ssDNA polymer are required for validation of the model.

FECs would be highly informative for the reader: Are the FECs reversible? Does the final FEC depend on the force used to flow the protein?

4) The experimental approach to measure intersegmental protein-protein interactions is not direct and may lead to misinterpretations about the protein behavior at low forces. The authors aimed to measure intersegmental transfer on a dsDNA molecule hold at a constant force of 80 pN. As the authors point out in the manuscript, above 60 pN dsDNA undergoes a cooperative transition (overstretching) due to strand peeling from free ends and internal melting of base pairs (King et al., 2013). The authors assumed that under these conditions, they have two parallel, unpaired ssDNA strands in between the beads. However, this may not be the case; the overstretching transition would lead to a much more 'complex' DNA molecule with peeled free ends under no tension (the free end in their construct (Figure 1) plus additional ends as a result of possible nicks), ssDNA under tension and unpaired 'parallel' ssDNA. Extrapolations to zero force conditions of the results obtained from the interactions of A3G with dsDNA partially denatured with force are, at least, questionable.

It may not be necessary to study intersegmental protein-protein interactions to support the main points of this work. If authors want to address intersegmental interactions, experiments at low forces (<10 pN) would be more convincing.

Reviewer #3:

Summary:

I believe the system studied is of great interest. I believe understanding the binding and diffusion properties of A3G in detail are important. I believe the experiments presented were conducted properly to the best of their capabilities. But my enthusiasm at the moment is low because the measurements are very indirect with respect to the conclusions they wish to draw. They seek to construct a detailed molecular model of the binding and diffusion of individual and oligomers of A3G on ssDNA. However, they do not measure individual A3G molecules or complexes, but the mean activity of very many of them on a single ssDNA tether. They also do not directly measure protein motion or diffusion.

Further, I have very significant concerns about the data analysis methods which they rely on to draw their core conclusions. 1) They do not seem to be considering the polymer properties of the ssDNA+bound-protein adequately. Oddly do not consider the possibility of the persistence length increasing upon binding and this would seem to explain their data more simply than multiple binding modes which are not directly observed. 2) They use a simple force independent Bell model fit to find a 'distance to the transition state' which they interpret as a contraction of the free ssDNA to the protein bound structure. Especially considering the extremely wide force range (10-80 pN) they are fitting, for sure they need to use a Dudko-style force dependent method and the d_b_ value they consider is not likely relevant to the actual structure of the ssDNA+A3G transition state.

My major concerns in more detail:

They perform a force-independent Bell model fit to the k_on_ vs force data to obtain a single 'distance to the transition state', d_B_, value. This distance describes the change in extension between (1) the unbound state, i.e., a stretched length of ssDNA, whose ext is very force dependent over the 10 – 80 pN fitting range, and (2) the protein+ssDNA transition state. Since the extension of the initial state (1) is very force dependent, the Bell model fit is very inappropriate. Even if a simple exponential fit can reasonably describe the data, the interpretation of the d_b_ = 0.088 nm value is challenging, and generally we would expect it to be dominated by the δ extension of the stretching initial ssDNA with changing force, and not any details of the transition state. But it is exactly the transition state they are interested in. In this case, they need to perform more complex force dependent fitting as described by Dudko et al. (2008 and beyond) and used by many groups in similar situations. They need to integrate the δ extension over force in the Bell exponent. To do that they need to explicitly model the extension of both the unbound tethered ssDNA and the transition state. To do that, they need a better hypothesis of what the structure of the transition state might be. But this is good, because that is in part what they are ultimately after. Can they do a better job guessing the structure and comparing hypothetical models to their data with a correct force dependent fitting method? If so they could get a lot more information about the ssDNA+protein bound structure.

I also have a major concern that they are not fully considering the polymer properties of the bound ssDNA+protein. Without knowing the subunit structure, e.g., the number of ssDNA bound and their geometry, it is unclear what the subunit contour and persistence lengths are. As they point out, the persistence length probably shouldn't decrease. Oddly they don't consider the possibility of it increasing. But it seems very likely to increase with increasing subunit size. It seems that if the protein binds a few nt without wrapping, we would expect an increase in extension, not a decrease. This is because the overall tether contour length stays nearly the same, e.g., two nt captured by a single protein creates a new polymer subunit approximately twice the size, but also reduces the number of subunits from 2 to 1, and the hence the overall tether contour length stays the same. But the persistence length likely increases with the growing subunit size. Oligomerization interactions would seem to increase the persistence length further. This would increase the overall tether extension at a given force not decrease it (see Equation 1). And indeed, this simple model also explains why the difference in extension/nt decreases with increasing force. This is how Equation 1 behaves with a change in persistence length but fixed contour length: at high force you approach the contour length no matter the persistence length, and the biggest difference occurs at low force. A factor of two increase in persistence length gives a factor of 7 decrease in the δ ext from 10 to 80 pN, similar to the factor of 4 they observe. In fact, their Figure 3C looks nearly exactly like what you expect from plotting the difference between two FJC models with the same contour length but a change in persistence length of ~2. This simpler polymer thinking does not require considering multiple binding modes, as they do, which are not supported by any direct observations. However, they still shouldn't ignore the possibility that wrapping or bending of the ssDNA would decrease the protein+ssDNA subunit size compared the free DNA. Adding these countering effects together could give a net small change in extension.

[Editors’ note: what now follows is the decision letter after the authors submitted for further consideration.]

Thank you for resubmitting your work entitled "HIV restriction factor APOBEC3G binds in multiple steps and conformations to search and deaminate single-stranded DNA" for further consideration by *eLife*. Your revised article has been evaluated by John Kuriyan (Senior Editor) and a Reviewing Editor.

The manuscript has been improved but there are some minor remaining issues that need to be addressed before acceptance, as outlined below:

Summary:

Understanding how Apobec3G (A3G) interacts with DNA may provide important clues to resolve the mystery of how it manages to efficiently deaminate the HIV genome. In the previous submission of this paper, Morse M. et al. used single-molecule force spectroscopy (optical tweezers) to measure the binding and unbinding kinetics of the restriction factor APOBEC3 (A3G) under increasing mechanical tensions. The authors seek to construct a detailed molecular model of the binding and diffusion of individual molecules and oligomers of A3G on ssDNA. One of the major claims is that A3G binds single-stranded DNA (ssDNA) in multiple steps and in two distinct conformations.

In this revised paper, the authors have made a great effort responding to our previous comments and suggestions. Most importantly, the authors present data at very low forces, which is a more physiological situation. More rigorous assessment of the polymeric properties of the ssDNA-AG3 complex proves the authors conclusion that AG3 changes the persistence length and also yields the transition distance associated with formation of the A3G-ssDNA loops. Another interesting aspect of new data is comparison of slow (equilibrium) FECs where the AG3-ssDNA complexes/loops have a chance to equilibrate with fast FECs where the protein was pre-bound either at low or at high force. The new experimental and theoretical evidence provides stronger support to their conclusions.

Overall, the reviewers agree that this this will be a valuable *eLife* paper especially given the care and rigor with which they've handled investigating the protein binding modes coupled to the polymer properties.

Revisions requested:

The reviewers felt that after addition of the new data without condensing the overall text, the paper became difficult to read with a lot of mixed presentation of experiments with speculative discussion and very complex, if not convoluted arguments. A simpler, clearer presentation of these results can be made. We would suggest the authors streamline the paper, move non-essential results to supplementary, and distinguish better between the more speculative findings. For example, what is the low force 'looped' configuration? The authors say we should just call this a 'looped' configuration for convenience, but then proceed as if it is literally a looped configuration, connecting to the putative multiple independent ssDNA binding domains which could bind and bring together disparate sections of ssDNA. This is confusing to the reader.

Another example of confusing analysis: In the Discussion the authors state: "If both domains are bound simultaneously, this will necessarily constrict the ssDNA substrate, resulting in the decrease in ssDNA extension that we observe." Why? They have already noted that binding of a protein to ssDNA can have the effect of simply increasing the persistence rate. Why would increasing the amount of protein bound (in the absence of a wrapping mechanism as for e.g., *E. coli* SSB or nucleosomes) not simply further increase the persistence length? Following that, again, a 1 nm loop doesn't sound like a loop at all, compared to the zero force data (Figure 5G) which are very plausibly loops.

The authors also need to clarify the following points:

1) Figure 1C and Figure 5G show the force-extension curves of the ssDNA-A3G complex after incubation at low force: Figure 1C shows a 'smooth' extension increase and the effect of protein binding is noticeable even at 80 pN. However, Figure 5G shows an abrupt saw- tooth pattern, where (in most cases) the extension of the protein-DNA complex equals that of ssDNA at F~40 pN. Why are these differences between the two plots (considering experimental conditions are similar)?

2) Figure 2E shows the change in extension during A3G dissociation. Because wild-type A3G does not dissociate below 50 pN, does data shown in Figure 2E correspond to the mutant? Please, clarify this point in the figure.

3) The authors aimed to measure the transfer of A3G between ssDNA segments (intersegmental transfer) on a dsDNA molecule hold at a constant force of 80pN. They assume that under these conditions, tension induces the formation of two parallel, unpaired ssDNA strands in between the beads. However, the overstretching transition may lead to much more 'complex' DNA structures, which calls into question the conclusions obtained from these experiments. Please clarify.

4) Subsection “A3G binding conformations” paragraph one. It is stated that the maximal decrease in extension is observed at 20 pN. However, Figure 2B shows greater changes in extension for 10 pN. Why is this?

5) How do the results shown in Figure 4B agree with the FECs shown in Figure 1C and 1D and with the dissociation rates reported in Figure 2?

6) Subsection “A3G binding conformations” paragraph three. It is not clear how the authors calculate the energy term EL~3 kBT.

Is this value supposed to explain the low binding affinity of the CTD for ssDNA?

How does the value of EL explain the remarkable mechanical stability of the “large loops structures” (Discussion paragraph one) that characterize the FECs shown in Figure 5G?

7) Please give citations for the polymer models, e.g., Equation 10.

---

## [Author Response]

[Editors’ note: the author responses to the first round of peer review follow.]

Reviewer #1:

Understanding how Apobec3G (A3G) interacts with DNA may provide important clues to resolve the mystery of how it manages to efficiently delaminate HIV genome. In this work, the authors applied optical tweezers single-molecule technique to interrogate the A3G binding to ssDNA tethered between two beads. This follows the authors previous study that used DNA melted by application of force (2x DNA). Here, the ssDNA was produced by the T7-endonuclease activity, which resulted in a single ssDNA molecule between the two beads and allowed the authors to carry out binding experiments at broader range of forces and to build a detailed energy landscape of A3G-DNA interactions in two different conformational states. Additionally, comparing the results of 1x and 2x DNA the authors were able to conclude that A3G can also bridge the two DNA molecules, or at low or zero forces to stabilize the ssDNA loops.Overall, the experiments are expertly performed, the authors' conclusions are interesting and are in agreement with previously published data by the authors and others (in particular with the FRET and AFM data). The model of two binding modes and the conclusion that the enzymatically "active" state of A3G corresponds to the contracted conformation of A3G-DNA complex where the catalytic CTD is fully engaged is reasonable.

We thank the reviewer for their positive assessment of both our experiments and our interpretation of A3G function. Through further experiments and analysis, we believe we fully addressed the following comments.

Major issues that need to be addressed1) The authors' argument that physical effect of A3G binding must be a reduction in the DNA counter length is plausible and quite likely. One cannot dismiss, however, another possibility, namely that the change in the elastic stretching modulus upon A3G binding may also play a role.

As suggested by reviewer 2, we now explicitly show force extension curves for ssDNA bound by A3G. This in turn allows us to more explicitly define how A3G changes ssDNA’s polymer properties. We discuss in more detail in the manuscript, but in short, A3G actually increase persistence length which increases extension especially at low force. Furthermore, we observe no evidence that elastic modulus increases (we actually see a decrease in certain circumstances, though this likely an artifact of the A3G mediated loops formed at low forces being disrupted at high forces). Thus, the only way the ssDNA length can be decreased is due to contour length reduction. This is consistent with the ssDNA loops mediated by A3G binding, which we show more evidence for using new low force experiments.

2) Quantitative description of the A3G binding energy landscape depends on the A3G DNA binding site, which the authors estimate to be 18 nt based on the overall size of the protein. This estimation needs an experimental confirmation.

The most direct way to experimentally measure binding site size is observing the binding of a protein to a short fixed length of ssDNA, which are exactly the experiments performed by the Lyubchenko group using AFM, which we reference. Measuring binding site size directly using optical tweezers would require identifying individual binding events. This is possible for proteins such as *E. coli* SSB in which a single tetramer can decrease ssDNA extension on the order of tens of nm. However, the binding of single A3G monomers results in extension changes on the order of a single nm and we are unable to resolve this over the inherent force noise of a bead in an optical trap. However, we now detail the kinetics of A3G binding based on the equilibrium ssDNA extension change due to A3G binding (Figure 5) and show that A3G’s binding site size is on the order of 15 nt. Additionally, using data from new experiments, we are now able to quantify the inter-conformational kinetics in manner that is not sensitive to binding site size.

Reviewer #2:

Morse M. et al. use single-molecule force spectroscopy (optical tweezers) to measure the binding and unbinding kinetics of the restriction factor APOBEC3 (A3G) under increasing mechanical tensions. One of the major claims is that A3G binds single-stranded DNA (ssDNA) in multiple steps and in two distinct conformations. Overall, the paper is well written and data seems consistent. However, the theoretical model and the major conclusions of the paper are based on experiments performed at moderate or high tensions on ssDNA (10-80 pN) and protein concentrations favoring oligomerization, which calls into question whether the results could be extrapolated to physiological conditions.

We thank the reviewer for their constructive comments, which we addressed by performing two new sets of experiments. First, we captured a series of force-extension curves of A3G saturated ssDNA that show the exact manner in which A3G binds ssDNA (and changes its polymer properties) depends on applied force. Second, we use a dsDNA/ssDNA hybrid construct to measure the binding of A3G to ssDNA at lower forces and in higher salt concentrations, allowing us to measure how ssDNA secondary structure impacts A3G binding. This new information allows us to address the comments below.

Main points:1) It is not clear why the authors did not perform experiments at tensions F<10 pN (which are more physiologically relevant). One of our major concerns is that at low tensions (<10 pN), the overall structure (organization) of ssDNA is governed by its large entropic elasticity and its propensity to form secondary structure: secondary structure may locally inhibit/control A3G binding to ssDNA (i.e. making it difficult to form oligomers) and entropic elasticity may additionally modulate binding modes and kinetics. Although the theoretical model includes the effects of entropic elasticity of the ssDNA through the FJC model, the putative effects of secondary structure on protein binding were not experimentally tested nor modeled. The role of the secondary structure of ssDNA would be even more relevant in experimental conditions with higher salt concentrations because increasing concentrations of Na^+^ and Mg^2+^ are known to strongly stabilize the secondary structure of nucleic acids. The authors observed that binding and oligomerization is robust under high ionic strength conditions (subsection “Force dependence of A3G binding kinetics”, Figure 2—figure supplement 1) and claimed that the behaviors observed in their study should be observed under physiological conditions and in vitro biochemical assays. This is arguable because they performed all the experiments at tensions that prevent the formation of secondary structure, assuming no role of this phenomenon on protein binging/unbinding kinetics. Therefore, extrapolation of their model to zero force conditions (physiologically relevant) may not be entirely correct. The authors should comment on why did not conduct experiments at low forces and what the effect of secondary maybe on results and conclusions.

We did not originally include lower force data in our analysis for two reasons. First, working with a fully single stranded DNA template limits the minimum tension that can be stably achieved. As the reviewer notes, at low forces ssDNA forms secondary structure, resulting in equilibrium extended lengths much less than that predicted by a polymer model such as the FJC. Since the entire construct is ssDNA, once compaction begins, the tethering beads will approach one another making measurements <5 pN impossible. Second, the presence of secondary structure complicates A3G binding such that these results may not be directly comparable to higher forces. However, as the reviewer also notes, these interactions do occur under physiological conditions, and are worth probing. To this end, we performed a new series of experiments utilizing a modified experimental method (see new Figure 5).

We first resolve the technical limitation by using a hybrid dsDNA/ssDNA construct in which 1000 nts of ssDNA is flanked on both sides by dsDNA handles that are attached to the beads. The long persistence length of the dsDNA allows us to relieve nearly all tension on the ssDNA and approach zero force without the beads coming into contact with the ssDNA or each other. We then characterize the behavior of ssDNA itself (i.e. how much does secondary structure compact ssDNA) at these low forces so that these effects can be disentangled from further effects caused by A3G binding. These results significantly extend the range of conditions over which we perform our experiments and further support our presented model of A3G function.

2) The theoretical model assumes some points that should be reconsidered:A) The authors assume full protein coverage of the ssDNA at all tensions. However, at high forces the ratio between the measured Kd(f) and the experimental protein concentration, c, [Kd(f) / c = 10nM/50 nM = 0.2], indicates that at least 20% of the ssDNA chain would not be covered. This, in turn, would imply errors of this order or higher for the reported parameters. Could the 'putative' differences in protein coverage reduce the differences measured in the extension change per bound protein? (Less coverage would lead to less change in extension). The authors should consider including in their model protein coverage at its force dependence.Protein coverage could be determined by the comparison of the binding energy with the elastic energy difference between the covered and uncovered segment ΔG(elastic, covered)-ΔG(elastic, uncovered). Elastic energy at constant force would be given by ΔGelastic=∫x(f)df, with x(f) the force-extension curve for each of the two states (covered and uncovered).

We thank the reviewer for this suggestion, which we now explicitly discuss in the manuscript. Based on our calculated kinetics, the ssDNA should be >85% saturated even at high forces at 50 nM A3G. We now explicitly show this correction (Figure 3S), which is smaller than the presented error bars and does not meaningfully change the force dependence of the extension change. Additionally, we also now show force extension curves (see point below) of ssDNA with 200 nM A3G to ensure saturation and these values agree within error. We use these A3G saturated ssDNA curves in comparison with bare ssDNA to calculate the stretching energy ΔG required for both chains to reach a given force F. This ΔG helps calculate the required equilibration time due to binding at a given constant force and in turn better represent the force dependence of these kinetics as presented in the updated Figure 4.

B) Equation 7 assumes that the persistence length of the A3G-ssDNA polymer equals the persistence length of naked-ssDNA. However, this may not be the case; the persistence length may be of the order of the size of the A3G monomer (or higher, for cooperative binding).

We use the new FEC curves to show that persistence length does increase with A3G binding (similar to our previous measurements with A3H), which we now take into account. However, since increased persistence length increases ssDNA extension, the measured reduction in contour length is even more pronounced now.

3) Equation 7 is supposed to fit the force-extension curve (FEC) of the A3G-ssDNA polymer. However, the authors do not show the final FECs of the protein-DNA complexes nor their corresponding fits with the model. Fits to the FECs of the A3G-ssDNA polymer are required for validation of the model.FECs would be highly informative for the reader: Are the FECs reversible? Does the final FEC depend on the force used to flow the protein?

We did not previously show FECs for A3G bound ssDNA as stretching experiments since there is not in fact a unique FEC. Instead, the FEC depends on the initial binding force (which determines A3G’s binding state) and pulling rate (which determines how much A3G is able to re-equilibrate as force changes). We now show such FECs in the updated Figure 1. In short, the FEC starting at low force shows greatly reduced contour length while the FEC starting at high force shows increased persistence length but minor reduction in contour length (as such, the FECs are not reversible). This is consistent with increasing large ssDNA loops being stabilized by A3G at low forces and being pulled apart at high forces. We also obtain an equilibrium FEC by utilizing a slow pulling rate, which reproduces the force dependent change in extension observed in our previous constant force measurements. As for the flow force, while introducing the protein into the sample creates an artificial hydrodynamic force on the trapped bead, the optical tweezers instrument is calibrated with this effect taken into account, so the reported force is true tension on the DNA.

4) The experimental approach to measure intersegmental protein-protein interactions is not direct and may lead to misinterpretations about the protein behavior at low forces. The authors aimed to measure intersegmental transfer on a dsDNA molecule hold at a constant force of 80 pN. As the authors point out in the manuscript, above 60 pN dsDNA undergoes a cooperative transition (overstretching) due to strand peeling from free ends and internal melting of base pairs (King et al., 2013). The authors assumed that under these conditions, they have two parallel, unpaired ssDNA strands in between the beads. However, this may not be the case; the overstretching transition would lead to a much more 'complex' DNA molecule with peeled free ends under no tension (the free end in their construct (Figure 1) plus additional ends as a result of possible nicks), ssDNA under tension and unpaired 'parallel' ssDNA. Extrapolations to zero force conditions of the results obtained from the interactions of A3G with dsDNA partially denatured with force are, at least, questionable.It may not be necessary to study intersegmental protein-protein interactions to support the main points of this work. If authors want to address intersegmental interactions, experiments at low forces (<10 pN) would be more convincing.

As the reviewer suggests, the overstretched dsDNA experiments are not directly comparable to the current experiments in this paper. We mainly discuss these previously published results to explain the quantitative differences (especially in regards to extension change amplitude) to these new results. We hypothesize that these deviations from an idealized two parallel strand ssDNA are exactly the reason why we see divergent results. For instance, if bound A3G prevented peeling from the free ends which normally occurs during overstretching, it would drastically increase the elastic stiffness of the DNA resulting in a large decrease in extension (Figure 3C dotted line). This section has been rewritten to properly reflect the limitations of this older work. As such the strongest evidence for ssDNA looping is seen in lower force data, including increased extension change at lower forces, two step contraction at 10 pN, and the new experiments performed below 10 pN in the presence of ssDNA secondary structure, as suggested by the reviewer.

Reviewer #3:

Summary:I believe the system studied is of great interest. I believe understanding the binding and diffusion properties of A3G in detail are important. I believe the experiments presented were conducted properly to the best of their capabilities. But my enthusiasm at the moment is low because the measurements are very indirect with respect to the conclusions they wish to draw. They seek to construct a detailed molecular model of the binding and diffusion of individual and oligomers of A3G on ssDNA. However, they do not measure individual A3G molecules or complexes, but the mean activity of very many of them on a single ssDNA tether. They also do not directly measure protein motion or diffusion.

We appreciate the reviewer recognizing the importance of the system being studied and quality of the conducted experiments. While we cannot characterize the binding of A3G to ssDNA using force spectroscopy at a single protein level due to the small associated change in ssDNA extension, we do believe that the ensemble behavior of a fixed number of proteins on the substrate does reveal important features of A3G function.

Further, I have very significant concerns about the data analysis methods which they rely on to draw their core conclusions. 1) They do not seem to be considering the polymer properties of the ssDNA+bound-protein adequately. Oddly do not consider the possibility of the persistence length increasing upon binding and this would seem to explain their data more simply than multiple binding modes which are not directly observed.My major concerns in more detail:They perform a force-independent Bell model fit to the k_on_ vs force data to obtain a single 'distance to the transition state', d_B_, value. This distance describes the change in extension between (1) the unbound state, i.e., a stretched length of ssDNA, whose ext is very force dependent over the 10 – 80 pN fitting range, and (2) the protein+ssDNA transition state. Since the extension of the initial state (1) is very force dependent, the Bell model fit is very inappropriate. Even if a simple exponential fit can reasonably describe the data, the interpretation of the d_b_ = 0.088 nm value is challenging, and generally we would expect it to be dominated by the δ extension of the stretching initial ssDNA with changing force, and not any details of the transition state. But it is exactly the transition state they are interested in. In this case, they need to perform more complex force dependent fitting as described by Dudko et al. (2008 and beyond) and used by many groups in similar situations. They need to integrate the δ extension over force in the Bell exponent. To do that they need to explicitly model the extension of both the unbound tethered ssDNA and the transition state. To do that, they need a better hypothesis of what the structure of the transition state might be. But this is good, because that is in part what they are ultimately after. Can they do a better job guessing the structure and comparing hypothetical models to their data with a correct force dependent fitting method? If so they could get a lot more information about the ssDNA+protein bound structure.I also have a major concern that they are not fully considering the polymer properties of the bound ssDNA+protein. Without knowing the subunit structure, e.g., the number of ssDNA bound and their geometry, it is unclear what the subunit contour and persistence lengths are. As they point out, the persistence length probably shouldn't decrease. Oddly they don't consider the possibility of it increasing. But it seems very likely to increase with increasing subunit size. It seems that if the protein binds a few nt without wrapping, we would expect an increase in extension, not a decrease. This is because the overall tether contour length stays nearly the same, e.g., two nt captured by a single protein creates a new polymer subunit approximately twice the size, but also reduces the number of subunits from 2 to 1, and the hence the overall tether contour length stays the same. But the persistence length likely increases with the growing subunit size. Oligomerization interactions would seem to increase the persistence length further. This would increase the overall tether extension at a given force not decrease it (see Equation 1). And indeed, this simple model also explains why the difference in extension/nt decreases with increasing force. This is how Equation 1 behaves with a change in persistence length but fixed contour length: at high force you approach the contour length no matter the persistence length, and the biggest difference occurs at low force. A factor of two increase in persistence length gives a factor of 7 decrease in the δ ext from 10 to 80 pN, similar to the factor of 4 they observe. In fact, their Figure 3C looks nearly exactly like what you expect from plotting the difference between two FJC models with the same contour length but a change in persistence length of ~2. This simpler polymer thinking does not require considering multiple binding modes, as they do, which are not supported by any direct observations. However, they still shouldn't ignore the possibility that wrapping or bending of the ssDNA would decrease the protein+ssDNA subunit size compared the free DNA. Adding these countering effects together could give a net small change in extension.

Using newly shown force-extension curves, we now measure a ~2x increase in ssDNA persistence length due to the binding of A3G. This effect is similar to one we have previously measured for Apobec3H (Feng et al., 2018). This, however, does not explain the force dependent change in extension caused by A3G binding. Increased persistence length strictly increases ssDNA extension at all forces, such that decreased contour length must be the primary cause of ssDNA contraction (we see no evidence of an increase in elastic modulus and its effect would be negligible anyway at the low forces where we see the greatest effect). However, while a fixed contour length reduction could be partially offset by an increased persistence length at high forces, at lower forces the persistence length change would dominate, and we would observe increased ssDNA extension (see new Figure 1C, red curve). Thus, taking into account persistence length change, the reduction in contour length must be even greater in value than we previously calculated.

We have additional experimental evidence of these different binding states. Our previously shown force jump experiments (now Figure 4B and C) demonstrated that the ssDNA-A3G complex would re-equilibrate to a new state when the applied tension is altered (which is inconsistent with a single binding state). We now additionally show force extension curves (Figure 1C) showing that while the high force binding state has a small decrease in contour length mostly offset by increased persistence length, low force binding results in excess loop formation such that a larger contour length reduction dominates. We also utilize a slow pulling rate to maintain equilibrium binding for the ssDNA-A3G complex, which agrees with the previously shown constant force binding experiments (Figure 3A), and explicitly show that this X(F) behavior cannot be reproduced by a simple polymer model (Figure 1D). Finally, using a new binding construct that more closely approximates the physiological conditions of viral replication (lower force, higher salt, and increased ssDNA secondary structure formation), we perform a new series of experiments (Figure 5) where we directly detect A3G-mediated ssDNA loops, which are preferentially formed at low forces.

2) They use a simple force independent Bell model fit to find a 'distance to the transition state' which they interpret as a contraction of the free ssDNA to the protein bound structure. Especially considering the extremely wide force range (10-80 pN) they are fitting, for sure they need to use a Dudko-style force dependent method and the d_b_ value they consider is not likely relevant to the actual structure of the ssDNA+A3G transition state.

With the additional information available from the new experiments, we are now able to perform a similar mode of analysis as suggested by the reviewer. We note, however, there are a couple of key differences between our data and the types of experiments discussed in Dudko et. al, PNAS, 2008, which include examples of observing individual hairpins unzipping and proteins unfolding as applied force is increased. First, all our experiments which we quantitatively analyze are performed at a constant force or with a slow rate of extension change to ensure equilibrium is maintained. Thus, our results do not have an explicit dependency on the pulling rate (dx/dt) or rate of force change (dF/dt). Second, we are not observing individual events which are then used to construct a histogram of events over a range of forces. Instead, we observe many events along a single substrate which reach an equilibrium along a measurable timescale. We do, however, utilize the general concept of integrating over the change in extension over force to calculate a force dependent stretching energy that can be inserted into the Bell exponent to better represt the energetic barrier between the different binding states. In particular the newly presented force extension curves allow us to more explicitly measure the X(F) relationship of the different ssDNA-A3G states, allowing for calculation of the stretching energy offset ΔG (see Figure 4). For example, by integrating over the force-dependent Δextension between bare ssDNA and A3G saturated ssDNA at equilibrium, we are able to fit to our previous constant force binding experiments. Since the X(F) is fully defined for both states, the main fitting parameter is binding site size N, which we show is ~15 nt, in agreement with previously published AFM studies. We also apply this analysis to our force jump experiments.

[Editors' note: the author responses to the re-review follow.]

Revisions requested:The reviewers felt that after addition of the new data without condensing the overall text, the paper became difficult to read with a lot of mixed presentation of experiments with speculative discussion and very complex, if not convoluted arguments. A simpler, clearer presentation of these results can be made. We would suggest the authors streamline the paper, move non-essential results to supplementary, and distinguish better between the more speculative findings.

Multiple figure panels have been moved to supplement figures to better focus on the key results. We have removed some speculation from the Results section, leaving only what is necessary to motivate our experiments.

For example, what is the low force 'looped' configuration? The authors say we should just call this a 'looped' configuration for convenience, but then proceed as if it is literally a looped configuration, connecting to the putative multiple independent ssDNA binding domains which could bind and bring together disparate sections of ssDNA. This is confusing to the reader.

For clarity, this conformation is now referred to as “compacting” in the result section, as this is what we directly observe. We do believe the synthesis of all our results strongly supports that this compaction is the result of A3G forming variable size loops on the flexible ssDNA substrate, which we address in the Discussion.

Another example of confusing analysis: In the Discussion the authors state: "If both domains are bound simultaneously, this will necessarily constrict the ssDNA substrate, resulting in the decrease in ssDNA extension that we observe." Why? They have already noted that binding of a protein to ssDNA can have the effect of simply increasing the persistence rate. Why would increasing the amount of protein bound (in the absence of a wrapping mechanism as for e.g., *E. coli* SSB or nucleosomes) not simply further increase the persistence length? Following that, again, a 1 nm loop doesn't sound like a loop at all, compared to the zero force data (Figure 5G) which are very plausibly loops.

Thank you for pointing this out. Our previous statement was erroneous. We now specify that the additional protein binding places additional constraints (not constriction) on the ssDNA conformation. As you state, these constraints can indeed result in ssDNA extension by local straightening, which we now explicitly address. However, in our experiments, the dominant effect is ssDNA compaction.

The authors also need to clarify the following points:1) Figure 1C and Figure 5G show the force-extension curves of the ssDNA-A3G complex after incubation at low force: Figure 1C shows a 'smooth' extension increase and the effect of protein binding is noticeable even at 80 pN. However, Figure 5G shows an abrupt saw- tooth pattern, where (in most cases) the extension of the protein-DNA complex equals that of ssDNA at F~40 pN. Why are these differences between the two plots (considering experimental conditions are similar)?

The difference between these two curves are due to the different DNA constructs. Figure 1 uses a fully ssDNA 8.1 knt substrate while Figure 5 uses a hybrid substrate with 1 knt of ssDNA flanked by 5.5 kbp of dsDNA. We now explicitly explain why these curves are not identical: “The presence of dsDNA handles allows the ssDNA region to be fully unextended, allowing for potentially more A3G mediated compaction, as opposed to the fully ssDNA substrate which always has non-zero extension” and “These curves also appear more jagged than the stretch curves of the fully ssDNA substrate (Figure 1C) because the A3G-ssDNA complex is initially equilibrated at near zero force allowing further compaction, the ssDNA is 8X shorter (resulting in less averaging over length), and the stiff dsDNA handles allow for a greater drop in force after an extension increase event.” We also note in the caption to Figure 5E that the initial force is “near zero force so that the ssDNA region is completely unextended”

2) Figure 2E shows the change in extension during A3G dissociation. Because wild-type A3G does not dissociate below 50 pN, does data shown in Figure 2E correspond to the mutant? Please, clarify this point in the figure.

The figure legend is shared between Figure 2 A and B and D and E as these plots show the same data with different y-axes. This is now made explicit in the figure caption. As such, WT dissociation is only shown for F≥50 pN with the lower force data showing FW mutant dissociation.

3) The authors aimed to measure the transfer of A3G between ssDNA segments (intersegmental transfer) on a dsDNA molecule hold at a constant force of 80pN. They assume that under these conditions, tension induces the formation of two parallel, unpaired ssDNA strands in between the beads. However, the overstretching transition may lead to much more 'complex' DNA structures, which calls into question the conclusions obtained from these experiments. Please clarify.

The experiments in question are from an older study, which does indeed have complicating factors not present in the new experiments presented. We mainly attempt to explain deviations between our new and old results. As such the figure for these older experiments have been moved to supplemental as the new experimental techniques, which avoid such complicating factors, should take precedent.

4) Subsection “A3G binding conformations” paragraph one. It is stated that the maximal decrease in extension is observed at 20 pN. However, Figure 2B shows greater changes in extension for 10 pN. Why is this?

To be more clear in our experimental design, we now state we equilibrated the ssDNA A3G complex at 20 pN prior to the force jump as “A3G significantly compacts ssDNA but we do not observe the secondary compaction that we attribute to larger loop formation”. The maximal decrease in extension is indeed a more complicated question, as noted by the reviewer. Figure 3A presents the equilibrium change in extension due to A3G binding as data points representing the equilibrium change from Figure 2B or as lines from the equilibrium FECs from Figure 1C. Depending on the initial force, the change in extension due to A3G binding differs significantly, as shown as green and red lines on the same figure. Taken together, on average the curves show a maximal change in extension between 10 and 20 pN. In addition to removing the statement that 20 pN is the maximal extension, we also now clarify the Figure 3A caption accordingly.

5) How do the results shown in Figure 4B agree with the FECs shown in Figure 1C and 1D and with the dissociation rates reported in Figure 2?

The rate reported in Figure 2F k^off^ ~ 0.01 s^-1^ is the off rate for complete A3G monomer dissociation. It is much slower than the relaxation rate of the system presented in Figure 4B and C, which range from ~0.1 s^-1^ to 1 s^-1^. The latter rate reflects the much faster kinetics of the A3G-ssDNA complex transition between two modes of ssDNA binding, which occur without dissociation. This is now mentioned in the caption to Figure 4.

6) Subsection “A3G binding conformations” paragraph three. It is not clear how the authors calculate the energy term EL~3 kBT.Is this value supposed to explain the low binding affinity of the CTD for ssDNA?How does the value of EL explain the remarkable mechanical stability of the “large loops structures” (Discussion paragraph one) that characterize the FECs shown in Figure 5G?

We now clarify “In the absence of applied force, the much faster rate of A3G compaction results in this state being greatly favored (~90% occupancy). This implies the compacted state is favored by an energy term EL on the order of 3 kBT, assuming the two states occupancies can be calculated via Boltzmann distribution”. Our model does suppose that the energy is associated with the binding of the CTD. Presumably, these large loop structures are only stabilized by multiple A3G monomers since the binding occurs in the complete absence of an extending force.

7) Please give citations for the polymer models, e.g., Equation 10.

Equation 10 is just Equation 1 (which is cited) into which we substituted a reduced contour length and increased persistence length, which is now clearly stated in text.